



# Optimal closed-loop wake steering, Part 2: Diurnal cycle atmospheric boundary layer conditions

Michael F. Howland[1,2], Aditya S. Ghate[3,4], Jesús Bas Quesada[5], Juan José Pena Martínez[5], Wei Zhong[5], Felipe Palou Larrañaga[5], Sanjiva K. Lele[3], and John O. Dabiri[2,6]

[1]Civil and Environmental Engineering, Massachusetts Institute of Technology, Cambridge, MA 02139
[2]Graduate Aerospace Laboratories (GALCIT), California Institute of Technology, Pasadena, CA 91125
[3]Department of Astronautics and Aeronautics, Stanford University, Stanford, CA 94305
[4]NASA Ames Research Center, Moffet Field, CA 94035
[5]Siemens Gamesa Renewable Energy Innovation & Technology, Sarriguren, Navarra, Spain, 31621
[6]Department of Mechanical and Civil Engineering, California Institute of Technology, Pasadena, CA 91125

**Correspondence:** Michael F. Howland (mhowland@mit.edu)

**Abstract.** The magnitude of wake interactions between individual wind turbines depends on the atmospheric stability. We investigate strategies for wake loss mitigation through the use of closed-loop wake steering using large eddy simulations of the diurnal cycle, where variations in the surface heat flux in time modify the atmospheric stability, wind speed and direction, shear, turbulence, and other atmospheric boundary layer flow (ABL) features. The closed-loop wake steering control methodology developed in Part 1 (Howland *et al., Wind Energy Science*, 2020, 5, 1315-1338) is implemented in an eight turbine wind farm in large eddy simulations of the diurnal cycle. The optimal yaw misalignment set-points depend on the wind direction, which varies in time during the diurnal cycle. To improve the application of wake steering control in transient ABL conditions with an evolving mean flow state, we develop a regression-based wind direction forecast method. We compare the closed-loop wake steering control methodology to baseline yaw aligned control and open-loop lookup table control for various selections of the yaw misalignment set-point update frequency, which dictates the balance between wind direction tracking and yaw activity. Closed-loop wake steering with set-point optimization under uncertainty results in higher collective energy production than both baseline yaw aligned control and open-loop lookup table control. The increase in wind farm energy production for closed- and open-loop wake steering control compared to baseline yaw aligned control, is 4.0–4.1% and 3.4–3.8%, respectively, with the range indicating variations in the energy increase results depending on the set-point update frequency. The primary energy increases through wake steering occur during stable ABL conditions. Open-loop lookup table control decreases energy production in the convective ABL conditions simulated, compared to baseline yaw aligned control, while closed-loop control increases energy production in convective conditions.



# 1 Introduction

Collective wind farm power maximization through wake steering control has demonstrated potential in large eddy simulations (LES) of idealized atmospheric boundary layer (ABL) conditions (Gebraad et al., 2016), wind tunnel experiments (Campagnolo et al., 2020), and in initial field experiments (Fleming et al., 2019; Howland et al., 2019; Doekemeijer et al., 2021). The primary approach of wake steering control has been open-loop, where a lookup table of model-optimal yaw misalignment set-points is constructed as a function of the incident wind direction, wind speed, and turbulence intensity (Fleming et al., 2019). The

set-points are optimized using a steady-state, physics-based wake model and applied at the wind farm in time based on an estimate of the incident wind conditions at the farm. However, several challenges arise in open-loop wake steering control, including time-varying ABL flow conditions with measurement uncertainty (Quick et al., 2017; Annoni et al., 2019) and wake model parameter uncertainty (Schreiber et al., 2019; Howland, 2021b), which may lead to a discrepancy between the optimal yaw misalignment set-points in the steady-state wake model and the true optimal yaw misalignment values which vary in time.

Recent studies have developed closed-loop control methodologies (Ciri et al., 2017; Campagnolo et al., 2020; Doekemeijer et al., 2020; Howland et al., 2020c) which improve wake steering performance in flow with evolving mean states by incorporating wind farm measurements to modify wind condition (Doekemeijer et al., 2020) and wake model parameter (Howland et al., 2020c) estimates. The reader is directed to Part 1 of this study (Howland et al., 2020c) for further motivation of closed-loop wake steering control. Doekemeijer et al. (2020) investigated the performance of a proposed closed-loop control methodol-

ogy in LES of the idealized neutral ABL with a prescribed time-varying wind direction. Howland et al. (2020c) evaluated the performance of closed-loop wake steering control in the conventionally neutral ABL, which is characterized by neutral stratification in the boundary layer capped by a stable free atmosphere (e.g. Allaerts and Meyers, 2015), with fixed boundary conditions. The occurrence of the conventionally neutral ABL is rare in practice since the flow in the boundary layer is generally affected by non-neutral atmospheric stability. While numerical investigations often isolate atmospheric stability to

characterize its effects (Abkar and Porté-Agel, 2015), the transition between states of stability influences the ABL structure (Basu et al., 2008b; Fitch et al., 2013) and affects wind farm performance (Abkar et al., 2016). In this study, we investigate the performance of the closed-loop wake steering control methodology developed in Part 1 (Howland et al., 2020c) in the stratified ABL with time-varying wind direction and atmospheric stability.

Wind conditions evolve over the diurnal cycle through modifications to the surface heat flux (Stull, 2012). The daytime ABL

is characterized by surface heating and convection, giving rise to enhanced mixing and turbulent kinetic energy. Convective rolls with elongated streamwise length scales are observed for the weakly convective ABL (Deardorff, 1972; Atkinson and Wu Zhang, 1996; Salesky et al., 2017). Conversely, the stratification in the nocturnal ABL suppresses vertical velocity fluctuations and limits the flow length scales (Sullivan et al., 2003). The stable ABL is characterized by enhanced wind speed and direction shear (Wyngaard, 2010) and subgeostrophic (or low-level) jets (Thorpe and Guymer, 1977). Stable ABL low-level

jets are generated, in part, by inertial oscillations induced by Coriolis forces (Van de Wiel et al., 2010). Through modifications of the structure of the ABL, stratification influences wind farm performance (Wharton and Lundquist, 2012b).



In some instances, wind farm efficiency is diminished in stable conditions, compared to convective (Barthelmie and Jensen, 2010). Other studies have identified increases in power during stable ABL operation (Wharton and Lundquist, 2012a). Differences in reported wind farm performance in stable ABL conditions may relate to site- and time-specific wind direction shear

(Sanchez Gomez and Lundquist, 2020; Howland et al., 2020d) or low-level jets (Gadde and Stevens, 2021). Wind turbines generally operate in time-varying yaw misalignment due to slowly reacting yaw control systems and control error (Fleming et al., 2014). The power production of a wind turbine in yaw misalignment depends on the incident velocity field (Howland et al., 2020d; Liew et al., 2020). Since the wind speed and direction variations over the rotor area depend on the atmospheric stability, the power-yaw relationship for a given wind turbine depends on the stability (Howland et al., 2020d), in addition to

the control system in use. While the overall wind turbine performance depends on the interaction between these effects, the influence of stability on wake recovery is more clear. Wakes recover faster in convective ABL conditions compared to stable or neutral (Iungo and Porté-Agel, 2014), and relatedly, the wake meandering is enhanced (Abkar and Porté-Agel, 2015). Provided slower wake recovery as a function of streamwise distance downwind of a wind turbine in stable ABL conditions, wake interactions are generally increased (Abkar et al., 2016). Overall, the potential for wake steering control to increase wind farm

power production is anticipated to be higher in stable conditions, and initial empirical results confirm this trend (Fleming et al., 2019).

Wake models parameterize the effects of ABL turbulence on the wake recovery through a prescribed wake spreading rate (Jensen, 1983). Since the wake recovery depends on the atmospheric stability (Abkar and Porté-Agel, 2015), the wake spreading coefficient should depend on the wind conditions. Niayifar and Porté-Agel (2016) proposed a model for the wake spreading

rate as a function of the turbulence intensity, but the formulation considers only neutral stability. Instead, we leverage closed-loop control (Howland et al., 2020c) to estimate the wake spreading rate using time-dependent wind farm measurements. Through closed-loop control, the yaw misalignment set-point optimization adapts to the estimated wake model parameters, which vary with atmospheric stability. We anticipate that the primary benefits of the proposed closed-loop control approach result from adapting the model used for set-point optimization to time-varying wind conditions.

The optimal wake steering strategy depends on the wind conditions, including the wind speed, wind direction, and atmospheric stability. With the effects of turbulent diffusion parameterized with the wake spreading rate, the wind farm flow is estimated using a steady-state wake model with prescribed wind conditions (e.g. Gebraad et al., 2016). Recent studies have extended yaw misalignment set-point optimization to consider wind condition variability and uncertainty about the mean state of yaw misalignment (Quick et al., 2017), wind direction (Rott et al., 2018), and joint yaw misalignment and wind direction

(Simley et al., 2020). Howland (2021b) extended methods for set-point optimization under uncertainty to consider wake model parameter uncertainty, and empirical improvements for open-loop and closed-loop control were demonstrated. Quick et al. (2020) estimated the expected value of wind farm power under wind condition uncertainty using polynomial chaos expansion and demonstrated that wind direction uncertainty was the primary uncertainty in determining model-optimal yaw set-points.

Beyond wind condition variations about a known mean state, the low-frequency mean state of the atmosphere evolves in time

due to mesoscale meteorological processes (e.g. Sanz Rodrigo et al., 2017a) and the diurnal cycle (Kumar et al., 2006; Fitch et al., 2013) and is challenging to forecast. Existing wind farm control reacts to low-pass filtered wind condition measurements





(e.g. Fleming et al., 2019). Since the optimal wind farm control strategy inherently depends on the transient atmospheric conditions, wake steering control based on a forecast of future wind conditions over a finite time horizon is anticipated to improve performance, rather than reacting to past data. Recently, Simley et al. (2021) demonstrated in idealized wake model

numerical experiments that perfect wind direction preview information slightly improves wake steering control. In this study, we develop a regression-based statistical methodology to forecast future wind direction over a prediction horizon of minutes. We focus on a horizon of minutes based on the timescales of turbine yaw motors. In our approach, the yaw set-points are optimized using the wake model and the wind direction prediction, rather than the low-pass filtered historical wind direction data. The performance of wake steering control in transient ABL conditions also depends on the yaw misalignment update

frequency (Kanev, 2020), which represents a balance between yaw duty (frequency of yaw motor motions) and reacting to flow features of certain length and time scales. In this study, we compare the performance of closed-loop control to open-loop lookup table control for several yaw misalignment update frequency selections.

The set of findings presented here demonstrate the utility of closed-loop wake steering control in more realistic ABL conditions, with time-varying wind direction, wind speed, and atmospheric stability. This paper represents Part 2 of the closed-loop

wake steering control study presented by Part 1 (Howland et al., 2020c). The technical details associated with the model-based wake steering control are detailed in Part 1. Given recent advances in the literature, some methods are updated in this study, and the updates are described in §2. The diurnal cycle ABL case is described in §3 and the results are presented in §4. There are several appendices to provide supporting technical information. The wind direction forecast algorithm is in Appendix A. The diurnal cycle code validation is presented in Appendix B. Appendix C discusses the initialization of the LES cases for repro-

ducible numerical experiments of wind farm control. Finally, the lookup table construction, for open-loop control, is discussed in Appendix D.

## 2 Model-based closed-loop wake steering control methodology updates

The model-based closed-loop wake steering control methodology used in this study is presented in Section 2 of Howland et al. (2020c). Since the publication of Part 1, there have been several additional studies in the literature with improvements to wake

steering control methodologies. The updates to the wake steering methodology proposed in Part 1 are introduced in this section.

Several studies have investigated the superposition of individual wind turbine wakes in engineering wake models. Zong and Porté-Agel (2020) propose a momentum conserving superposition methodology under assumptions of uniform, steady inflow and negligible turbulent transport. Various wake superposition methodologies are investigated for the application of closed-loop control with parameter estimation by Howland and Dabiri (2021), which demonstrated that momentum conserving and mod-

ified linear superposition (Niayifar and Porté-Agel, 2016) perform similarly, while linear superposition (Lissaman, 1979) has degraded predictive accuracy. However, since the momentum conserving superposition (Zong and Porté-Agel, 2020), requires iterations, it is more computationally expensive than modified linear superposition. Therefore, modified linear superposition (Niayifar and Porté-Agel, 2016) is used in this study (more details are provided in Howland and Dabiri (2021)). The secondary steering model proposed by Howland and Dabiri (2021) is also used.



The power production of a yaw misaligned turbine depends on the incident flow field (Liew et al., 2020; Howland et al., 2020d). Howland et al. (2020d) developed a blade-element model which predicts the power production of a wind turbine in yaw misalignment given an incident ABL flow and validated the model with utility-scale turbine data operating under various wind speed and direction shear profiles and yaw misalignments. Since the present LES uses non-rotational actuator disk modeling (ADM), the blade element approach is not a representative model. Instead, we use the cosine model, $\hat{P}(\gamma_s) = \hat{P}(\gamma = 0) \cdot \cos^{P_p}(\gamma_s)$, where $P_p$ is a tuned empirical parameter. The $P_p$ exponent depends on the time-varying inflow. Additional inaccuracies arise in the cosine model since the power production as a function of the yaw misalignment is not generally symmetric in non-uniform flow (Howland et al., 2020d; Doekemeijer et al., 2021). Numerical experiments in Part 1 (Howland et al., 2020c) demonstrated that underestimating $P_p$ leads to poor wake steering performance. We select $P_p = 2.5$ for the particular ADM used in this study based on empirical tuning to LES of the conventionally neutral ABL (Howland and Dabiri, 2021). Since the main purpose of the present study is to characterize the performance of open- and closed-loop methodologies with a shared wake model, we do not dynamically adapt $P_p$ in the closed-loop method in this study. Future work should either use a blade element model to predict the power-yaw relationship for a rotating wind turbine model (Howland et al., 2020d) or adapt $P_p$ depending on the incident flow conditions for a non-rotational model.

Part 1 (Howland et al., 2020c) utilized deterministic programming to optimize the yaw misalignment set-points for fixed incident wind speed and direction. In this study, the yaw misalignment set-points are optimized using stochastic programming under wind condition (Quick et al., 2017) and model parameter uncertainty (Howland, 2021b). The deterministic and stochastic (optimization under uncertainty, OUU) programming approaches to yaw set-point optimization will be compared. Since the ADM used in this study has fixed $C_T$ and $C_p$ as a function of the wind speed, the wind direction is the primary factor influencing the yaw set-points (Quick et al., 2020). We therefore consider variations in wind direction $\alpha$ only. The yaw set-points are optimized at each control update step with period $T$. At current time $t$, the goal of the set-point optimization is to find the optimal yaw misalignment angles for time window $t$ through $t + T$. The yaw set-point optimization is given by

$$\gamma_s^*(\alpha, \boldsymbol{\psi}) = \underset{\boldsymbol{\gamma}_s}{\mathrm{argmax}}\, \mathbb{E}\left[\mathcal{G}(\alpha, \boldsymbol{\psi}, \boldsymbol{\gamma}_s)\right], \tag{1}$$

where $\mathcal{G}(\alpha, \boldsymbol{\psi}, \boldsymbol{\gamma}_s)$ is the wind farm power production as a function of the wind direction $\alpha$, yaw misalignment set-point $\gamma_s$, and wake model parameters $\boldsymbol{\psi}$. In this study, the wake model parameters to be estimated are the wake spreading rate $k_w$ and the Gaussian wake proportionality constant $\sigma_0$ for each turbine in the wind farm (see Part 1, Howland et al., 2020c). The optimal yaw misalignment set-point is $\gamma_s^*$. The expected value of the power production is

$$\mathbb{E}\left[\mathcal{G}(\alpha, \boldsymbol{\psi}, \boldsymbol{\gamma}_s)\right] = \int \cdots \int f(\alpha) f(\boldsymbol{\psi}) \mathcal{G}(\alpha, \boldsymbol{\psi}, \boldsymbol{\gamma}_s) d\alpha, d\boldsymbol{\psi}. \tag{2}$$

The probability distributions are indicated by $f(\cdot)$. The probability distributions are estimated using the wind farm data collected over the window $t - T$ through $t$, with current time $t$. The mean wind direction estimate for the next period ($t$ through $t + T$) is indicated by $\hat{\alpha}$. The wind direction is assumed to be uniformly distributed between $\hat{\alpha} - \sigma_\alpha$ and $\hat{\alpha} + \sigma_\alpha$, where $\sigma_\alpha$ is the standard deviation in time of the wind direction measured over the interval $T$ with a sampling rate of $\Delta t$, the computational time step in LES. Other wind direction probability distributions may be considered in future work. Methods for estimating $\hat{\alpha}$





are discussed in §2.1. The model parameter probability distributions are estimated using the methodology proposed in Howland (2021b), although it is noted that $f(\psi)$ can be estimated using Bayesian uncertainty quantification in future work. Eq.
2 is approximated using numerical quadrature with the midpoint rule. The yaw set-points are optimized using Eq. 1, solved with gradient-based optimization (Howland et al., 2019). While gradient-based optimization of Eq. 1 may be affected by local extrema, the analytic gradient-based optimization enables real-time set-point optimization on the order of seconds for the eight turbine case considered here. Future work may consider the combination of gradient-free search algorithms and gradient-based optimization. In this study, closed-loop control cases with deterministic yaw set-point optimization are also performed. The
deterministic yaw set-point optimization is the method presented in Part 1, with deterministic wind directions and a single set of wake model parameters estimated using the ensemble Kalman filter (EnKF) (Evensen, 2003).

## 2.1 Statistical wind direction forecast

Existing wake steering control methodologies, including in Part 1, implement yaw misalignment angles based on the low-pass filtered measurements of the wind direction (see e.g. Fleming et al., 2019; Howland et al., 2020c). However, due to turbulent and
large-scale wind variations, the wind direction varies in time. Methods which react to previous low-pass filtered wind direction measurements may implement a suboptimal yaw misalignment strategy, depending on the future wind direction trajectory. A recent study using idealized wake model numerical experiments by Simley et al. (2021) demonstrated that using perfect preview wind direction measurements improves wake steering but using a preview based on a empirically fit cross-spectrum, between the wind direction measurements of two neighboring turbines, did not increase power over the standard method. The
empirically fit cross-spectrum model based wind direction prediction requires measurements of the wind direction by a wind turbine, MET mast, or LiDAR at an upwind location.

The goal of the optimization (Eq. 1) in closed-loop control is to estimate the optimal yaw set-point angles $\boldsymbol{\gamma}_s$ for the time window of $t$ to $t+T$, during which the yaw angles will be applied. In this study, we use a steady-state wake model for yaw set-point optimization which estimates the time averaged power production, based on time averaged wind conditions. With
perfect wind direction information, the yaw set-point optimization is performed at time $t$ with

$$\overline{\alpha} = \frac{1}{T} \int\limits_{t}^{t+T} \alpha(t')dt'. \tag{3}$$

We therefore focus on methods to forecast $\overline{\alpha}$. In this study, two methods are used to estimate $\overline{\alpha}$, with the estimate given by $\hat{\alpha}$. The standard approach (termed the filtered method) is $\hat{\alpha} = \frac{1}{T} \int_{t-T}^{t} \alpha(t')dt'$, which assumes that the low-pass moving average filtered wind direction is not changing. Note that some previous approaches use a first-order filter (e.g. Simley et al., 2020),
rather than a moving average filter, but we do not anticipate the results of the present study to be substantially different based on the particular wind direction filter used.

Here, we develop an alternative approach to estimate the future mean wind direction $\overline{\alpha}$ based on regression (termed predictive method). The wind direction forecast first uses wind direction data from $t-2T$ to $t$ to identify if the low frequency wind direction is stationary or varying. A linear regression model is fit to wind direction data from $t-2T$ to $t-T$. The regression





model is then used to predict the wind direction from $t - T$ to $t$. If the coefficient of determination ($R^2$) of the regression is above a set threshold value of $R_{min} = 0.2$ and the regression model has lower mean square error (MSE) than predicting the wind direction from $t - T$ to $t$ as $\frac{1}{T} \int_{t-2T}^{t-T} \alpha(t') dt'$, then the wind direction is considered to be varying, otherwise it is considered stationary. With the low-frequency wind direction determined to be varying, a second regression model is fit to wind direction data from $t - T$ to $t$. The future wind direction $\hat{\alpha}$ is then predicted using the second regression model at time $t + T/2$. If the

wind direction is considered stationary, it is estimated as $\hat{\alpha} = \frac{1}{T} \int_{t-T}^{t} \alpha(t') dt'$, the default filtered method. The full algorithm is presented in Appendix A in Figure A1 and Algorithm 1. While this method does not require external upwind wind direction measurements, it could be improved with additional upwind sensors. Closed-loop wake steering cases are performed in LES with both the filtered and predictive wind direction estimates.

## 3   Setup of large eddy simulations of the diurnal cycle

Large eddy simulations are performed using the open-source pseudo-spectral code *PadéOps*[1] (Ghate and Lele, 2017). The solver is introduced in detail in Part 1 (Howland et al., 2020c). The LES code has been previously used for simulations of the stable ABL (Ghate, 2018; Howland et al., 2020b). The code is validated for the simulation of the diurnal cycle through a comparison to the LES data of Kumar et al. (2006) in Appendix B. The equation for the transport of the filtered nondimensional potential temperature $\theta$ is given by

$$\frac{\partial \theta}{\partial t} + u_j \frac{\partial \theta}{\partial x_j} = \frac{\partial}{\partial x_j}\left(\frac{\nu_T}{P_r}\frac{\partial \theta}{\partial x_j}\right), \tag{4}$$

with velocity $u$, SGS heat flux with eddy viscosity $\nu_T$ and turbulent Prandtl number $P_r$. The wall model is constructed using the SURFFLUX1 algorithm (Basu et al., 2008a) to estimate friction velocity based on a prescribed surface heat flux. The computational domain size is $12 \times 4 \times 2$ kilometers with $480 \times 320 \times 320$ grid points in the $x$, $y$, and $z$ directions, respectively, with $z$ representing the wall-normal coordinate. We use the concurrent precursor methodology to simulate a finite wind farm

(see e.g. Munters et al., 2016; Howland et al., 2020a) with fringe regions (Nordström et al., 1999) in the last $25\%$ of the domain in the $x$ and $y$ horizontal directions.

    A representative diurnal cycle ABL is designed based on the study of Kumar et al. (2006). The geostrophic wind speed is fixed at $G = 8$ m/s and is in the positive $x$ direction. The wind speed is initialized with $u = G$ and $v = w = 0$ throughout the domain. The surface heat flux is prescribed following the time-varying profile shown in Figure 1(a). The full 24 hour diurnal

cycle is not simulated since the 12 period (Figure 1(a)) captures the stability transition of interest and for computational limitations. The domain is initialized with the potential temperature profile shown in Figure 1(b). The surface heat flux is initialized at $\overline{w'\theta'}_s = 0.05$ K·m/s, with positive and negative heat flux corresponding to surface heating and cooling, respectively. The convective ABL is run for one hour to remove startup transience before the wind farm control is initiated. A note on LES initialization for reproducible wind farm control numerical experiments is given in Appendix C.

---

[1]https://github.com/FPAL-Stanford-University/PadeOps

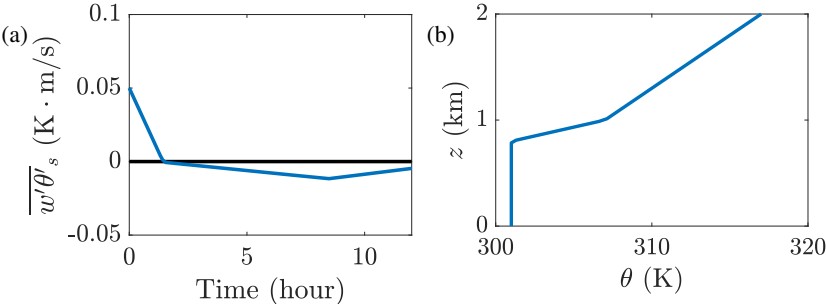

**Figure 1.** (a) Time-varying surface heat flux $\overline{w'\theta'}_s$. The simulation is initialized at time $t = 0$ corresponding to 18:00. Positive heat flux corresponds to surface heating and negative flux is cooling. (b) Initial potential temperature $\theta$ profile.

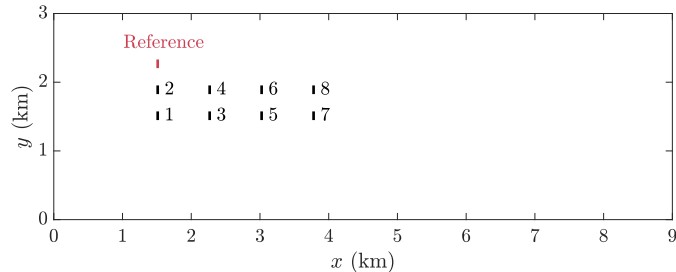

**Figure 2.** The wind farm layout considered in this study within the domain of interest (excluding the sections influenced by the fringe region). The reference turbine (shown in red) is used for power normalization and uses yaw alignment control for each case.

A nine turbine wind farm is located in the computational domain. The wind turbines are modeled using the ADM. The hub height is 100 meters and the rotor diameter is 126 meters. The coefficient of thrust is $C_T = 0.75$. The wind farm geometry is shown in Figure 2. Eight wind turbines are considered for wake steering control with one turbine used for reference. Given the initialization in the convective ABL, the wind direction in the ABL will initially be oriented in the positive $x$ direction (Figure 3(a)). As the surface heat flux becomes negative, the convective ABL will transition to a stable boundary layer. During

the transition, the reduced vertical mixing and inertial oscillations will result in an Ekman spiral, which is characterized by counter-clockwise turning of the wind from the geostrophic wind direction (parallel to isobars) to the surface wind direction (cross-isobaric). As a result, the mean wind direction at the wind turbine hub height will become positive (with the angle measured between the wind direction and the $x$ axis), as shown in Figure 3(a). A zoomed wind direction profile between hours 2 and 3 is shown in Figure 3(b) to show the turbulent variations. In summary, in the convective ABL, the flow will be

approximately in the positive $x$ direction, resulting in wake interactions along the columns of turbines. During the transition and stable regimes, the flow will be oriented with a positive angle, measured from the $x$ axis, and wake interactions will be along the farm diagonals (e.g. turbine 4 in the wake of turbine 1).

   The streamwise hub-height turbulence intensity in the inflow to the wind farm, computed from the concurrent precursor, is shown in Figure 3(c). The convective ABL is characterized by approximately 10% streamwise turbulence intensity. The



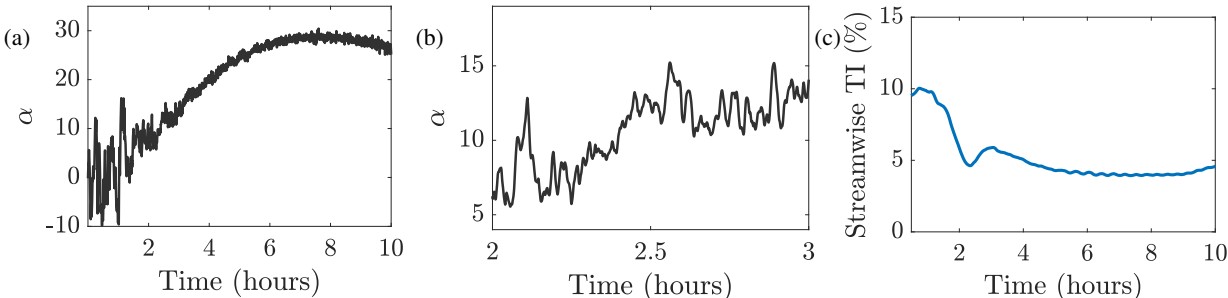

**Figure 3.** Diurnal cycle flow (a) hub height wind direction, (b) hub height wind direction zoomed to show variations between hours two and three, and (c) hub height turbulence intensity.

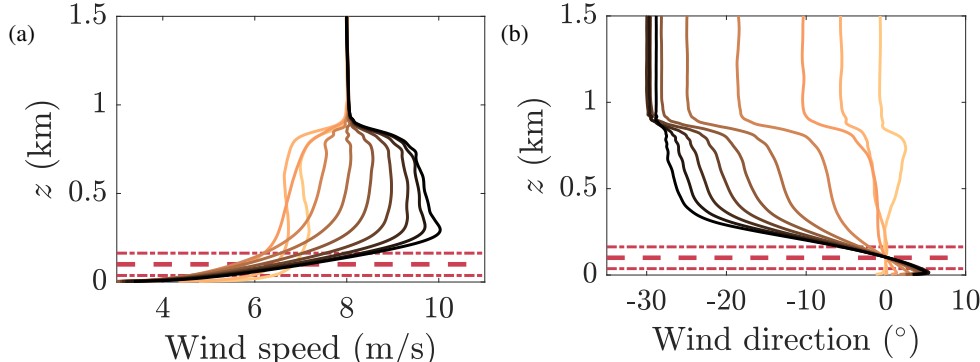

**Figure 4.** Diurnal cycle flow (a) wind speed and (b) wind direction, $\alpha(z) - \alpha(z_h)$, as a function of height $z$, where $z_h$ is the wind turbine hub height. The profiles are 30 minute averages at hourly intervals throughout the 12 hour simulation, with lighter colors near the initialization (unstable ABL) and darker colors corresponding to later times of the simulation (stable ABL). The horizontal dashed line corresponds to hub height and the horizontal dashed dotted lines are the rotor extents.

turbulence intensity decreases below $5\%$ during stable conditions. The incident wind speed profiles over the diurnal cycle are shown in Figure 4(a). The unstable wind speed has low shear above the near-wall region. As the flow transitions to nocturnal conditions, the shear across the rotor area is enhanced and a subgeostrophic jet emerges. Given the setup of the representative ABL used in this study, the maximum wind speed is above the rotor area. The wind direction as a function of height $\alpha(z) - \alpha(z_h)$ is shown in Figure 4(b). The wind direction change over the rotor area is minimal during the convective conditions and

is enhanced during stable conditions. The peak veer across the rotor area is approximately $15°$. The stable boundary layer wind direction variation as a function of height $z$ is consistent with Ekman turning (see e.g. Wyngaard, 2010).

     As the boundary layer transitions during the diurnal cycle, the structure of the turbulence will be modified, in addition to the mean wind profile changes. An instantaneous hub-height wind speed snapshot during convective conditions is shown in Figure 5(a) for wind turbines operating in baseline yaw aligned control. There are large-scale structures of high and low wind speed.

The wake meandering is qualitatively seen in the variations of the $y$ position of the wake velocity deficits as a function of $x$. The mean wind direction at hub height is in the positive $x$ direction during convective conditions. An instantaneous snapshot





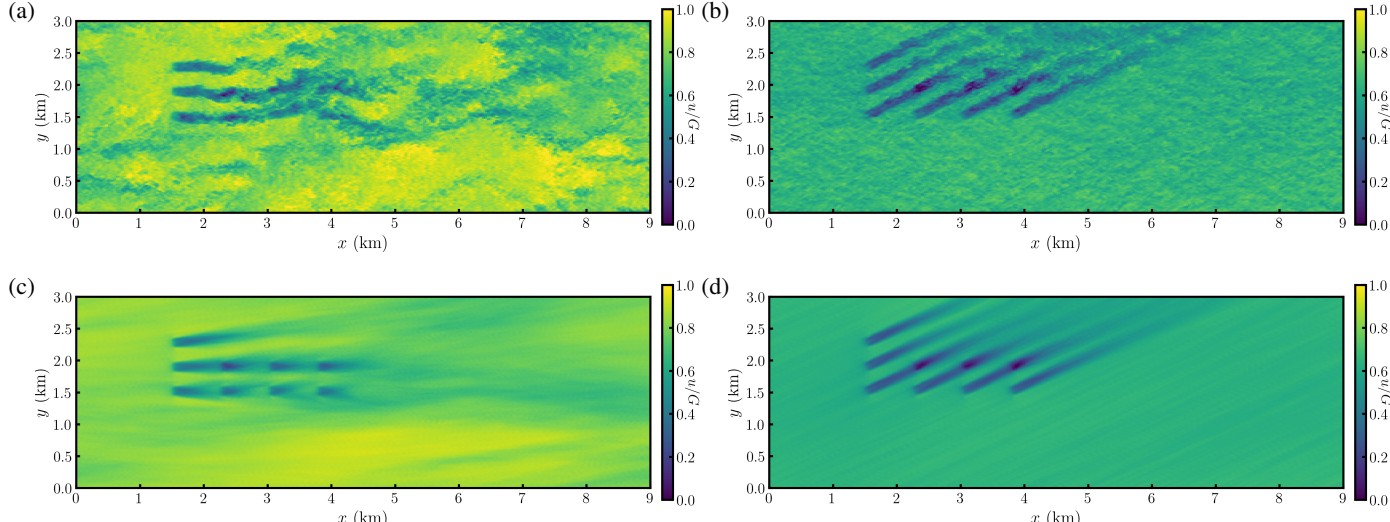

**Figure 5.** Hub height velocity during (a,c) unstable and (b,d) stable ABL conditions for the baseline yaw aligned control case. Instantaneous snapshots are shown in (a,b) and 10 minute moving averaged flow fields are shown in (c,d).

during stable conditions is shown in Figure 5(b). Compared to the convective conditions (Figure 5(a)), the stable flow field has diminished length scales and the wake meandering is reduced. The wind direction has also shifted, to approximately 20-30°, with respect to the $x$-axis (see Figure 3(a)). A 10-minute moving average of the instantaneous flow fields, sampled at a rate

of approximately 15 seconds, is taken for the convective and stable conditions, shown in Figure 5(c,d), respectively. The same timestep as the instantaneous snapshots is shown. The 10-minute moving average does not eliminate the heterogeneity from the convective ABL flow field. Longer time averages reduce flow field heterogeneity but also average over mean state transitions. Flow field heterogeneity can be physically modeled in future work (e.g. Starke et al., 2021; Martínez-Tossas et al., 2021). Conversely, the 10-minute moving average used for the stable conditions removed nearly all inflow heterogeneity. The time

averaged wake regions trailing the individual turbines are qualitatively different in the two atmospheric stability regimes. The effective wake diameters in the time averaged convective ABL are significantly larger than in stable conditions.

## 4   Wake steering results

In this section, wake steering control cases are run in the representative diurnal cycle simulation environment discussed in §3. The wake steering and yaw aligned control cases are run with a prescribed, fixed yaw update period of $T$. A baseline, yaw

aligned control case (Case A) is run for reference. As in Part 1, a basic yaw controller is used, such that the nacelle position of each turbine is updated to orient towards the mean wind direction measured at each local turbine, averaged over time $T$. We compare four wake steering control strategies. We consider one open-loop wake steering case and three closed-loop control cases, which differ only through their yaw set-point optimization methods.



Case D (D for deterministic) is closed-loop wake steering with deterministic yaw set-point optimization. The yaw set-points
are optimized with estimated wake model parameters using the EnKF and mean wind conditions prescribed as the average
conditions observed over previous time $T$. Case D-F (D for deterministic, F for wind direction forecasting) is closed-loop wake
steering with deterministic yaw set-point optimization which uses the wind direction forecast methodology. Comparing Cases
D and DF, differences will arise only from the wind direction used in the yaw set-point optimization. Case D uses the mean
wind direction measured over the previous time $T$ while Case D-F uses DirectionEstimation (Algorithm 1) to forecast the
wind direction over future time $T$. Case OUU-F uses optimization under uncertainty (OUU, see §2) and the wind direction
forecast methodology. For brevity, we do not include a case with OUU without the wind direction forecast. Case L uses
open-loop lookup table control. The lookup table synthesis is described in Appendix D. In §4.1, the power-yaw relationship
for the freestream turbines are presented. The performance of the various control strategies are compared in §4.2. The wake
model predictions are compared for closed- and open-loop control methodologies. The influence of the yaw update period is
considered in §4.3.

For the purpose of parsing the diurnal cycle results by atmospheric stability, we define the stable regime as $0 < L < 200$
(Van Wijk et al., 1990), where $L$ is the Obukhov length

$$L = -\frac{u^{*3}\theta_0}{\kappa g \overline{\theta' w'}_s}, \tag{5}$$

with friction velocity $u^*$, reference potential temperature $\theta_0$, von Karman constant $\kappa$, and gravitational acceleration $g$. For
$L < 0$, the flow is unstable or near neutral, while for $L > 200$ the flow is near neutral. Conditions of $L < 0$ and $L > 200$
are combined into unstable and stability transition periods. This stability characterization is qualitative and is used for the
interpretation of the results in the following sections.

### 4.1 Power-yaw relationship

The power productions of the leading two wind turbines in the array, turbines 1 and 2 (see layout in Figure 2), as a function
of their yaw misalignment with respect to the turbine-specific hub height wind direction, are shown in Figure 6. The results
are shown for a yaw update period of $T = 30$ minutes, and therefore, each data sample shown is a 30 minute average. Since
the wind direction changes as a function of time, the magnitude and sign of the model-optimal yaw misalignment set-points
will also change. Given the incident wind direction and wind farm geometry, turbine 2 will initially yaw misalign to benefit
turbine 4. With the wind direction shifting away from $0°$ with respect to the $x$-axis, there are no turbines downwind of turbine
2 (see Figure 5(b)) and its yaw misalignment set-point will become zero. Turbine 1 will continue to yaw misalign to benefit
either turbine 3 or turbine 4. The power ratios for the convective ABL are shown with open markers. Given the highly turbulent
convective ABL, the finite time averaged inflow wind to a given turbine may differ from the winds incident to the reference
turbine (see Figure 5). This effect is not accounted for in the cosine models, and is the primary cause for the significant spread
in the power ratios in convective conditions.

While there are a limited number of data samples for $\gamma > 0°$, the power ratio shown in Figure 6 appears asymmetric about
$\gamma = 0°$ during stable conditions (filled markers). Given the nocturnal wind speed and direction profiles shown in Figure 4,





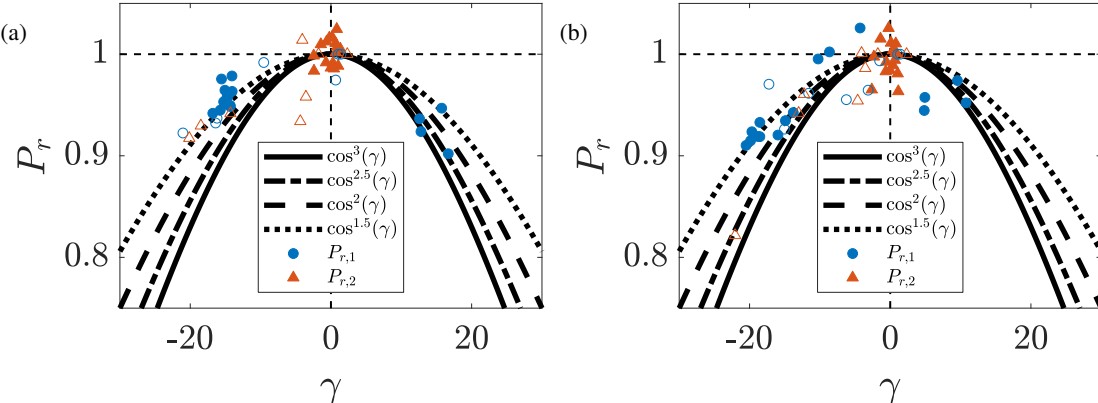

**Figure 6.** Power ratio $P_r = P_i/P_{\text{ref}}$ for turbines 1 and 2 (see layout in Figure 2) for (a) open-loop lookup table control and (b) Case D closed-loop control shown as a function of the realized yaw misalignment with respect to the hub height wind direction. Power ratios for stable atmospheric stability (Obukhov length $L < 200$) are shown with filled markers and hollow markers are unstable and stability transition periods. The power ratio is averaged for 30 minutes for each sample.

the power production for the yawed wind turbines will be asymmetric as a function of the sign of the yaw misalignment angle (Howland et al., 2020d). Considering a non-rotational actuator disk model representation of a wind turbine, the power production $P \propto (\boldsymbol{u} \cdot \hat{n})^3$, where $\boldsymbol{u}$ is the incident wind velocity vector and $\hat{n}$ is the unit vector normal to the rotor area. Given

the Ekman spiral, negative yaw misalignment, a clockwise rotation of the wind turbine viewed from above, will locally align the turbine above hub-height where the wind speed is larger than the hub-height speed (Figure 4(a)). Conversely, positive yaw misalignment will locally align the turbine below hub-height, where the wind speed is lower than hub-height speed.

The power ratio of turbine 1 for negative yaw misalignment is near the $\cos^{1.5}(\gamma)$ curve. Conversely, the data samples for positive yaw are generally between $\cos^{1.5}(\gamma)$ and $\cos^3(\gamma)$. In this study, the $P_p$ parameter for the simplified cosine power ratio

model $P_r = \cos^{P_p}(\gamma)$ was set to $P_p = 2.5$ based on previous tuning in conventionally neutral ABL conditions (see §2). Since the simplified cosine model is not the focus of the present study, and since the most accurate $P_p$ factor depends on the incident wind profiles and on the sign of $\gamma$, the value is not further tuned and is fixed at $P_p = 2.5$ for closed- and open-loop control cases. The results of Part 1 (Howland et al., 2020c) indicate that overestimating the power degradation as a function of the yaw misalignment angle is preferred for wake steering, compared to underestimation.

### 4.2 Comparison of control strategies

In this section, the various control strategies introduced in §4 are implemented in the diurnal cycle ABL flow with a fixed control update period of $T = 30$ minutes. We first investigate the influence of the wind direction estimation methodologies. The statistical wind direction forecast (§2.1) is compared to a wind direction estimate using a moving average filter with timescale $T$. The instantaneous $\alpha$ and mean $\alpha_T$ wind direction as a function of time, as measured by the reference wind

turbine, is shown in Figure 7. The mean wind direction estimates using a moving average and using the wind direction forecast

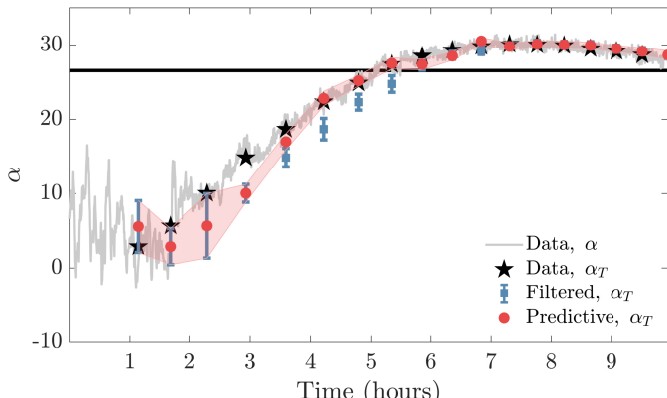

**Figure 7.** Comparison of the mean wind direction estimation methods to the measured instantaneous ($\alpha$) and mean ($\alpha_T$) wind direction data. A standard approach is shown where the low-pass filtered wind direction $\alpha_T$ is estimated through a moving average. The predictive method is shown where $\alpha_T$ is estimated using the proposed DirectionEstimation algorithm, described in Figure A1 and Algorithm 1. The horizontal black line corresponds to the wind direction of alignment between turbines 1 and 4.

methodology are shown, termed filtered $\alpha_T$ and predictive $\alpha_T$, respectively. The mean wind direction prediction methods have access to $\alpha(0:t)$, where $t$ is the current time, and predict $\alpha_T(t+T/2)$. In the limiting cases of high wind direction variability about a mean value (hours 0-3) or low mean wind direction changes in time (hours 6-10), the predictive methodology defaults to the same estimate as the filtered value. However, for periods of transitioning mean wind directions (hours 3-6), the predictive wind direction forecast more accurately estimates the mean wind direction for the future time horizon of length $T$. The mean absolute error (MAE) for the filtered and predictive methods for estimating $\alpha_T$ are 1.9° and 1.3°, respectively. The mean square error (MSE) for the filtered and predictive methods for estimating $\alpha_T$ are 6.0 and 3.7 (degrees squared), respectively.

Closed-loop wake steering control is implemented in the diurnal cycle ABL with deterministic yaw set-point optimization with the filtered (Case D) and predictive forecast (Case D-F) methodologies for the estimation of $\alpha_T$. Two separate LES cases are run with the only difference as the estimated mean wind direction ($\alpha_T$) provided to the yaw set-point optimizer. The yaw set-points for turbine 1 are shown in Figure 8(a) for the two cases. The realized yaw misalignment angles are shown in Figure 8(b). Since the initial conditions are fixed (the processor topology is also fixed, see Appendix C), during the initial four control update steps in which the filtered and predictive mean wind directions are the same (see Figure 7), the yaw misalignment values are identical. For step five and beyond, the estimates for the mean wind direction differ, resulting in a divergence of the yaw control approaches. The primary differences between the cases arise between hours 3.5 and 6, during which the mean wind direction transitions over the inflow angle of alignment between turbines 1 and 4. At this inflection point, the optimal yaw set-point angle will transition from positive to negative yaw. The predictive methodology estimates that the wind direction will transition to an angle greater than the inflection point, resulting in a negative yaw set-point, while the filtered methodology results in a positive yaw set-point. The positive yaw set-point, given the resulting trajectory of $\alpha$, results in wrong way steering that increases the wake losses at turbine 4.



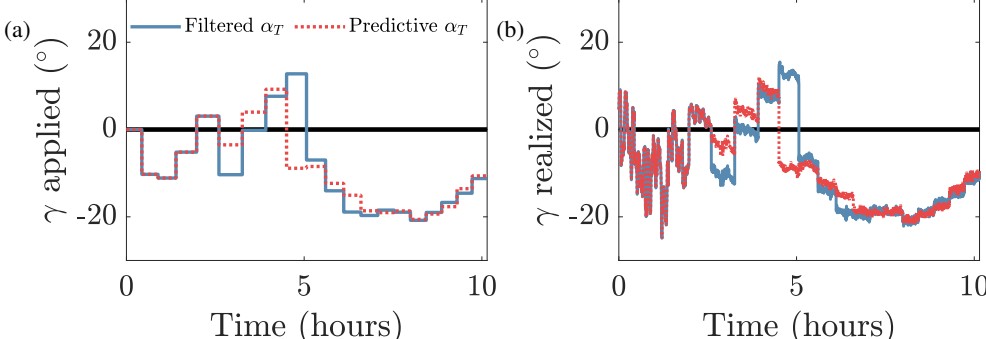

**Figure 8.** Yaw misalignment for turbine 1 (a) applied and (b) realized for the deterministic optimization methodology with filtered (Case D) and predictive forecast (Case D-F) wind direction-based yaw optimization.

The performance of each case is characterized using an energy ratio

$$E_r = \frac{\int_{t_1}^{t_2} \sum_{i=1}^{N_t} P_i(t)\mathrm{d}t}{\int_{t_1}^{t_2} \sum_{i=1}^{N_t} P_i^{\gamma_0}(t)\mathrm{d}t}, \tag{6}$$

which quantifies the wind farm performance compared to baseline yaw aligned control, indicated with $\gamma_0$, over time interval $t_1$ to $t_2$. The percent gain in energy through wake steering is $G = 100 \cdot (E_r - 1)$. We first focus on the time periods in which
the filtered and predictive wind direction methodologies differ (control update periods 6 through 9, approximately hours 3.5 to 6). The gain for this time period is $-0.1\%$ and $1.1\%$ for the filtered (Case D) and predictive (Case D-F) cases, respectively. As a result of the transitioning mean wind direction, reacting to the filtered history of wind direction results in the incorrect yaw misalignment direction, and therefore reduced energy production compared to baseline yaw aligned control. Conversely, the predictive wind direction methodology results in the appropriate yaw misalignment set-point direction and increases power
compared to baseline control. The energy gain $G$ for Cases D and DF for the full simulation are shown in Table 1. Overall, the wind direction forecast method increases the energy production using wake steering control in both atmospheric stability regimes, with the predominant energy improvements occurring during the time periods of hours 3.5 to 6, described above.

The energy gain resulting from the use of a wind direction forecast methodology in wake steering control is case specific and depends on several factors. First, the performance gain depends on the fidelity of the predictive methodology. In this study,
we have proposed a linear regression-based wind direction forecast which demonstrates empirical success in this application (Figure 7). For different ABL forcing, site-specific characteristics, or different update periods $T$, linear regression may not be sufficient, and other data-driven prediction approaches can be implemented in the framework outlined here (see Appendix A). Future work should consider nonlinear regression or more complex machine learning timeseries prediction methods. Further, the improvements herein predominantly stem from the occurrence of wind direction changes across the turbine alignment
inflow angle. The degree to which a wind direction forecast methodology improves overall wake steering performance will depend on the frequency of such occasions.



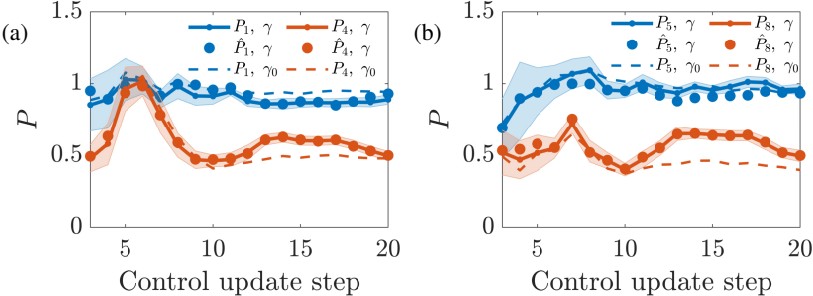

**Figure 9.** Power production results from the closed-loop control Case D and baseline yaw aligned control cases. The LES power data is given by $P$ and the wake model estimates are given by $\hat{P}$. The subscripts on the power denote the turbine number. The yaw misaligned and aligned cases are denoted by $\gamma$ and $\gamma_0$, respectively.

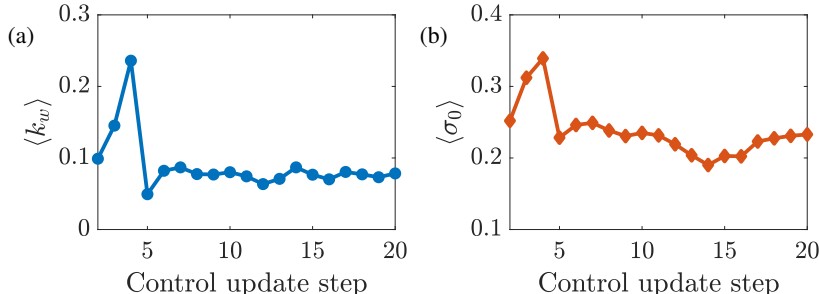

**Figure 10.** Estimated wake model parameters averaged over turbines 1, 3, and 5 as a function of the control update step. (a) Wake spreading coefficient $k_w$. (b) Gaussian wake proportionality constant $\sigma_0$.

In the closed-loop wake steering control approach proposed in Part 1 (Howland et al., 2020c), the wake model parameters $\psi$ are estimated at each control update step, with time increment $T$. The LES power production $P$, $\gamma$ as a function of the control update step for Case D is shown in Figure 9. In addition, the wake model power estimates $\hat{P}$, $\gamma$ for wake steering control and the LES power production for yaw aligned control $P$, $\gamma_0$, are shown. The power productions for the pair of turbines 1 and 4 are shown in Figure 9(a) and for turbines 5 and 8 in Figure 9(b). The wake model estimates for the power production of turbines 4 and 8 exhibit low error. Larger error arises in the prediction of upwind, freestream power production for turbines 1 and 5, given the simple cosine model (see discussion in §4.1). The power increase for the downwind turbines is more substantial in the stable regime (control update 9 and after). The estimated wake model parameters are shown in Figure 10. The parameters are averaged over the upwind turbines 1, 3, and 5. Both the wake spreading rate and the proportionality constant are reduced in stable atmospheric stability, compared to unstable conditions, as anticipated from the time averaged velocity fields (Figure 5).

Closed-loop wake steering control is implemented with optimization under uncertainty (see §2) and the wind direction forecast methodology in Case OUU-F. The energy gain results for Case OUU-F are shown in Table 1. Generally, set-point optimization under uncertainty (OUU) will reduce the magnitude of the peak yaw misalignment angles, especially near the inflow angle of alignment (see e.g. Quick et al., 2020). Given the high turbulence in the convective ABL, the wind direction





| Case | Deterministic | Deterministic, $\alpha$ forecast | OUU, $\alpha$ forecast | Lookup table (open-loop) |
|---|---|---|---|---|
| Label | Case D | Case D-F | Case OUU-F | Case L |
| Unstable & transition | −0.18% | 0.08% | 1.00% | −0.74% |
| Stable | 4.61% | 4.87% | 4.80% | 4.70% |
| Full simulation | 3.50% | 3.86% | 4.00% | 3.43% |

**Table 1.** Wind farm energy production increase compared to baseline yaw aligned control, $G = 100 \cdot (E_r - 1)$ with $E_r$ in Eq. 6. Cases with $\alpha$ forecast use the DirectionEstimation algorithm. Case OUU-F uses stochastic programming for yaw set-point optimization under uncertainty (OUU). The case with the highest overall wind farm energy production for a given time period is shown in green. The stable periods correspond to $0 < L < 200$ with unstable and transition times otherwise.

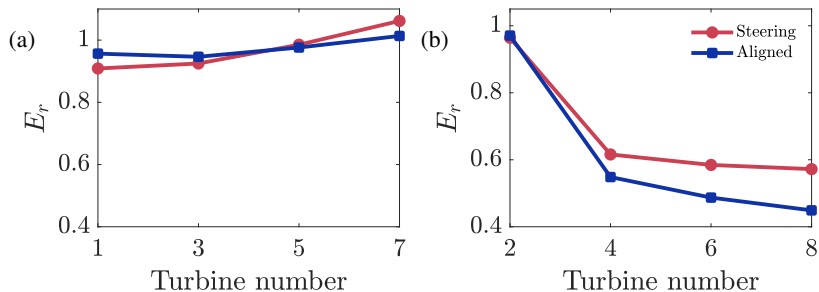

**Figure 11.** Diurnal cycle flow turbine-specific energy ratio $E_{r,i}$ (Eq. 7) for (a) the odd and (b) even rows. The odd row consists of turbines 1, 3, 5, and 7. The even row consists of turbines 2, 4, 6, and 8. The turbine layout is provided in Figure 2. Wake steering results from Case OUU-F are shown.

standard deviations are large (see Figure 7) and the yaw set-points will be reduced, compared to deterministic optimization. Case OUU-F has improved performance compared to Case D-F. The OUU (Case OUU-F) has improved performance as a result of increases in energy production during unstable and transition regimes. The energy production is slightly less for OUU in the stable regime. The energy ratio between times $t_1$ and $t_2$ for a given turbine is

$$E_{r,i} = \frac{\int_{t_1}^{t_2} P_i(t)\mathrm{d}t}{\int_{t_1}^{t_2} P^{\mathrm{ref}}(t)\mathrm{d}t}, \tag{7}$$


with the power production of the reference turbine given by $P^{\mathrm{ref}}$ (see Figure 2 for the layout). The reference turbine is used for normalization rather than $P_i^{\gamma 0}$ to maintain information in the turbine energy ratio metric $E_{r,i}$ about wake losses. The turbine energy ratios for Case OUU-F are shown in Figure 11. Small reductions in $E_{r,i}$ for yaw misaligned turbines (1, 3, and 5) result in large increases in energy ratios for the waked turbines (4, 6, and 8). Turbine 7 is not yaw misaligned during the simulation

and slightly outperforms the reference turbine, likely due to mean flow effects such as induction and blockage in the stable ABL (Segalini and Dahlberg, 2020).

Open-loop wake steering is implemented in the diurnal cycle ABL LES (Case L). The open-loop yaw misalignment lookup table synthesis is described in Appendix D. The yaw misalignment set-points and realized yaw values for turbines 1, 2, and 3





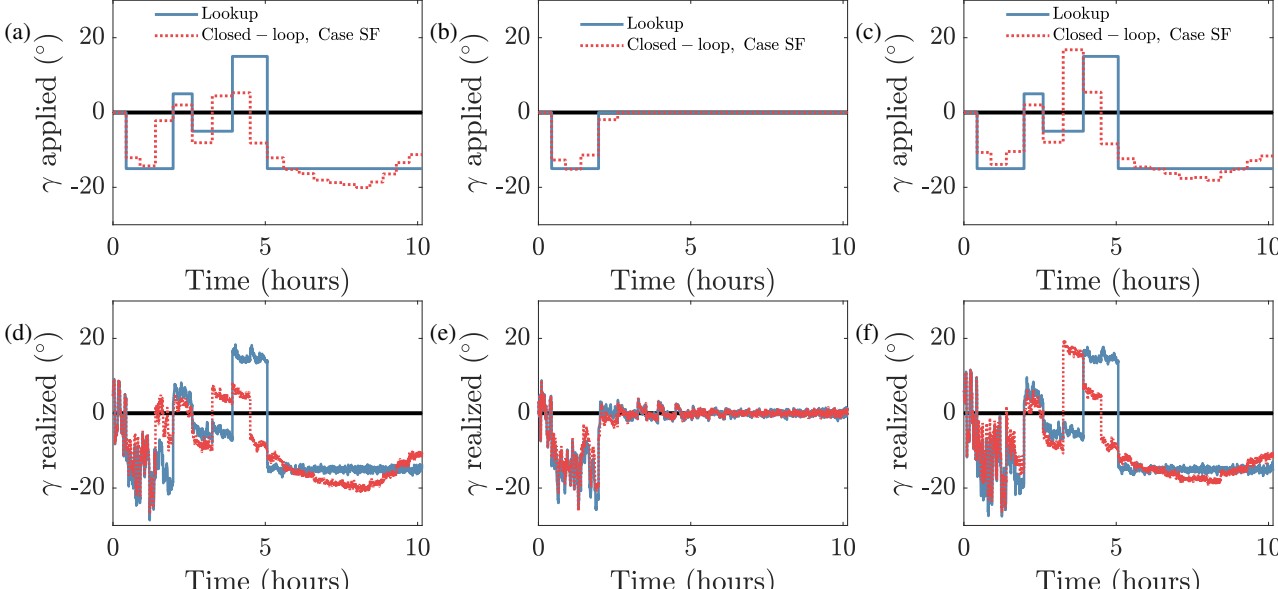

**Figure 12.** Comparison of the yaw misalignment set-point values as a function of time from the OUU closed-loop control (Case OUU-F) and from open-loop lookup table control (Case L) for turbines (a) 1, (b) 2, and (c) 3. (d-f) Same as (a-c) with realized yaw.

for closed-loop Case OUU-F and open-loop Case L are shown in Figure 12. The yaw misalignment set-points are qualitatively

similar in their approach but quantitatively differ. The differences between the closed- and open-loop yaw set-points are larger in the unstable and transition regimes of the simulation than the stable regime. The energy gains for the open-loop wake steering case are shown in Table 1. The lookup table control performance is similar to closed-loop control with deterministic set-point optimization but without the wind direction forecast method (Case D). Lookup table control has less energy production than baseline yaw aligned control for unstable and transition regimes, with a $0.74\%$ energy loss. For stable conditions, the open-

loop lookup table control has $4.70\%$ energy increase over baseline control. Overall, the open-loop control case has diminished performance compared to all closed-loop control cases.

The predictive performances of the open- and closed-loop control methodologies are assessed by comparing the power production predictions from the wake model to the LES power for stable atmospheric conditions. The row averaged power production is shown for upwind turbines, averaged over turbines 1, 3, and 5, and for downwind turbines, averaged over turbines

4, 6, and 8. The wake model power predictions from open-loop control, using the predefined wake model parameters depending on turbulence intensity in the inflow, are shown in Figure 13(a). The LES power production from the open-loop wake steering case (denoted with $\gamma$) is shown, in addition to the baseline yaw aligned control case (denoted with $\gamma_0$). The predefined wake model parameters result in significant predictive bias for the downwind waked turbines for both yaw aligned and wake steering control. The absolute errors are $0.146$ and $0.165$ for the yaw misaligned and yaw aligned wake model estimates, respectively.

The LES power production is compared to the closed-loop wake model estimates, where the wake model parameters are estimated using the ensemble Kalman filter, in Figure 13(b). The ensemble Kalman filter adapts the wake model parameters

WIND
ENERGY
SCIENCE
DISCUSSIONS

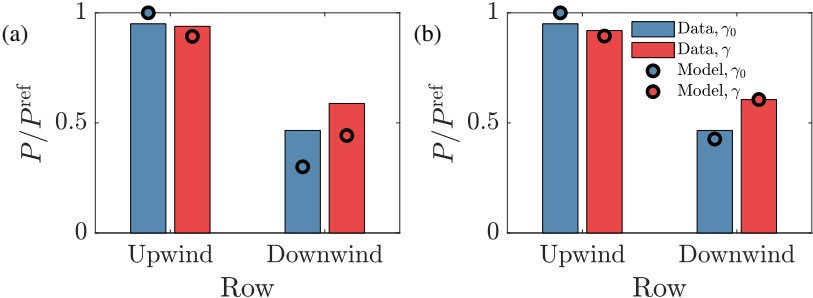

**Figure 13.** Comparison of LES row averaged power data and wake model predictions for the closed-loop (Case OUU-F) and lookup table (Case L) wind farm control methodologies for the stable stratification regime. (a) Lookup table control data and predictions. (b) Closed-loop control data and predictions. Baseline yaw aligned control results are indicated with $\gamma_0$ and wake steering results are indicated with $\gamma$. The upwind row is an average of turbines 1, 3, and 5. The downwind row is an average of turbines 4, 6, and 8.

to accurately estimate the wake steering power production. Since the closed-loop control LES power is used in the Kalman filter, this result is a wake model estimate (training data). Conversely, the wake model estimates for the power production in baseline yaw aligned control ($\gamma_0$) are predictions, since the Kalman filter does not use the power production from the separate yaw aligned LES case to estimate wake model parameters. The absolute errors are 0.0004 and 0.039 for the yaw misaligned and yaw aligned wake model estimates, respectively.

### 4.3 Comparison of yaw update periods

In this section, the sensitivity of the wind farm energy production for the various control cases to the yaw set-point update period $T$ is investigated. Baseline yaw aligned control and three wake steering cases previously described are implemented in LES of the diurnal cycle of the ABL with control update periods of $T = 30$ and $T = 15$ minutes. Case D is not repeated with $T = 15$ minutes in this section for brevity. Each case with a specified control update period represents an independent LES simulation. Again, all simulations are initialized from the same initial conditions. The energy gain, $G = 100 \cdot (E_r - 1)$ with $E_r$ in Eq. 6, with respect to baseline yaw aligned control Case A for $T = 30$ minutes is shown in Table 2. For yaw aligned control, decreasing $T$ will increase the frequency of updates wherein the nacelle position is updated according to the measured wind direction. It is therefore anticipated that reducing $T$ will increase the energy production in yaw aligned control (see e.g. Fleming et al., 2014), at the compromise of increased yaw duty (we do not account for the yaw motor energy consumption in this study). Table 2 demonstrates a $0.43\%$ increase in energy production for the baseline yaw aligned control with $T = 15$ minutes, compared to $T = 30$ minutes.

The wake steering control cases are implemented with $T = 15$ minutes. The yaw misalignment set-points and realized yaw for $T = 30$ and 15 minutes for open-loop lookup table control (Case L), closed-loop control with deterministic optimization (Case D-F), and closed-loop control with OUU (Case OUU-F) are shown in Figure 14. A lower control update period increases the variability in the yaw set-point values as the yaw control reacts to higher frequency timescales. Notably, the $T = 30$ minutes closed-loop control cases transition to negative yaw misalignment in the stable regime (around 5 hours) sooner than the faster



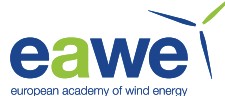
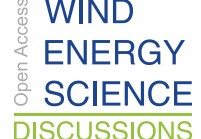

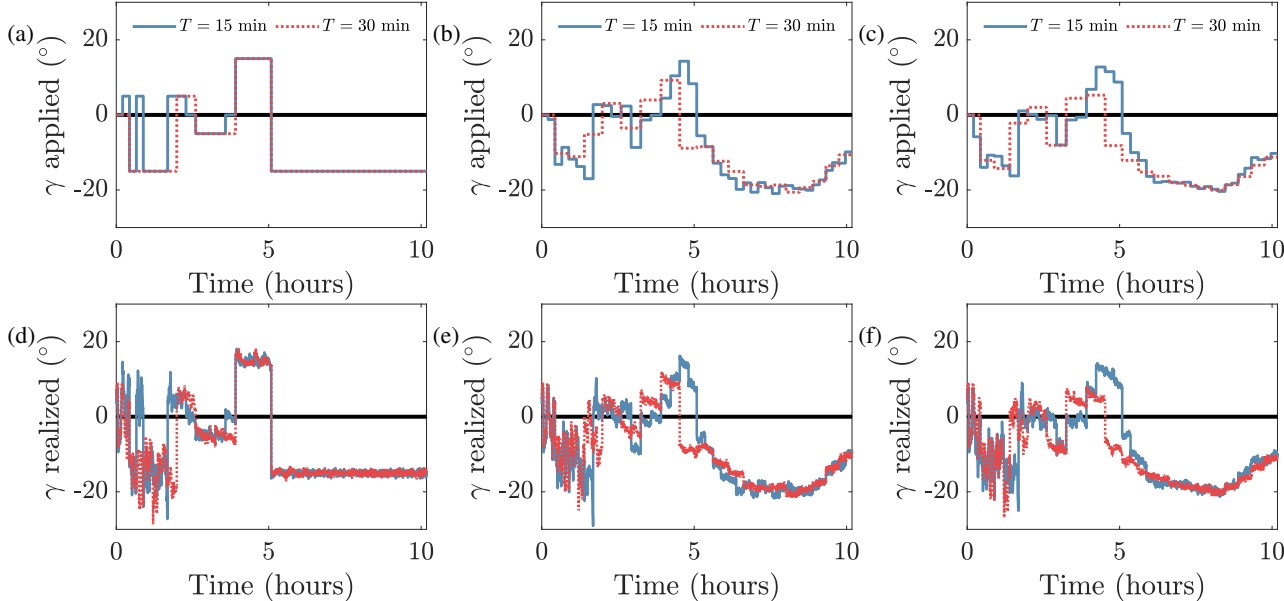

**Figure 14.** Comparison of the yaw misalignment set-point values as a function of time for turbine 1 from the (a) lookup table (Case L), (b) deterministic optimization (Case D-F), and (c) closed-loop control with optimization under uncertainty (Case OUU-F). (d-f) Same as (a-c) with realized yaw.

| Case | Baseline, $\gamma_0$ | Deterministic, $\alpha$ forecast | OUU, $\alpha$ forecast | Lookup table (open-loop) |
|---|---|---|---|---|
| Label | Case A | Case D-F | Case OUU-F | Case L |
| $T = 30$ min | – | 3.86% | 4.00% | 3.43% |
| $T = 15$ min | 0.43% | 3.82% | 4.14% | 3.83% |

**Table 2.** Wind farm energy production increase compared to baseline yaw aligned control with $T = 30$ min, $G = 100 \cdot (E_r - 1)$ with $E_r$ in Eq. 6. Cases with $\alpha$ forecast use the DirectionEstimation algorithm. The case with the highest overall wind farm energy production is shown in green. The full simulation period is considered, with stable, unstable, and transition regimes.

update frequency cases ($T = 15$ minutes). For $T = 15$ minutes, the wind direction forecast method defaults to the moving average filter for most time steps. For $T = 30$ minutes, the wind direction forecast results in negative yaw misalignment angles as the flow is transitioning across the inflow angle of turbine alignment (proactive), rather than after the transition has occurred (reactive).

The energy gain for each case with respect to the energy production in baseline yaw aligned control with $T = 30$ minutes is shown in Table 2. The highest energy production among all cases considered is optimization under uncertainty (Case OUU-F) with the wind direction forecast methodology and $T = 15$ minutes. The reduced yaw update period increases the energy production of closed-loop wake steering performed with OUU (Case OUU-F) while it slightly decreases the energy production





of closed-loop wake steering with deterministic set-point optimization (Case D-F). There are several factors which contribute to this result. The closed-loop control method estimates wake model parameters based on the average power measurements. For the closed-loop control methodology used here, the moving average for wind turbine power production uses the same timescale $T$ used by the control updates. Therefore, reductions in $T$ also reduce the time averaging length of the power production, which is used for parameter estimation. The reduction in $T$ increases the variability of the mean power data, by the central limit theorem. Reductions in $T$ may therefore lead to higher variability in the estimated wake model parameters. However, the averaging and control update timescale $T$ must be sufficiently small to adapt the wind farm control to the time-varying mean wind conditions. The selection of $T$ is a trade-off between these competing effects and may be site and ABL condition specific. While the optimal selection of $T$ is not the focus of this study, $T = 15$ minutes empirically demonstrates the highest overall energy production in these LES cases. The optimal update period should be investigated jointly with wind condition forecast methodologies. Future work should consider de-coupling the parameter estimate and control updates.

Contrary to deterministic set-point optimization, Case OUU-F, which utilizes set-point optimization under model parameter uncertainty, has improved performance with decreasing update periods $T$. This empirical result is also reproduced for unstable ABL conditions in Howland (2021b). Optimizing yaw misalignment set-points under a distribution of model parameters reduces the sensitivity to noise in the wind farm power production data. The effect of reducing the yaw update period for open-loop control is anticipated to be similar to baseline yaw aligned control, since the yaw set-points have been pre-defined in the lookup table. The energy production from open-loop control is increased by $0.4\%$ by reducing $T$ to $15$ minutes. For $T = 15$ minutes, open-loop lookup table control (case L) has a similar performance to closed-loop control with deterministic optimization (Case D-F). Closed-loop control with yaw set-point OUU (Case OUU-F) has the highest energy production for both yaw update periods and the highest overall energy production occurs with $T = 15$ minutes.

## 5   Conclusions

Closed-loop wake steering methodologies are investigated in a representative ABL with time-varying surface heat flux. The surface heat flux approximates the transitions which occur in the terrestrial diurnal cycle. Due to the variations in the surface heat flux, the character of the turbulence in the ABL is modified with a time-varying atmospheric stability. Convective ABL conditions, which are characterized by large-scale motions and high turbulence, result in enhanced wake mixing and reduced wake losses. Conversely, stable ABL conditions reduce the length-scales and intensity of the turbulence in the ABL and generally result in more substantial wake losses due to diminished wake diffusion. Existing methodologies for wake steering are open-loop, wherein a wake model with tuned wake spreading rates, which parameterize turbulence, is used to optimize the yaw misalignment set-points. The set-points are tabulated in a lookup table and applied based on measured wind conditions. In the closed-loop wake steering control methodology introduced in Part 1 (Howland et al., 2020c), and extended in this study, wind farm power production data is used to estimate wake model parameters, which vary in time, and the yaw misalignment set-points are optimized online.



The optimal yaw misalignment set-points depend on the incident wind direction. Rather than assuming that the future wind
direction will be equal to the low-pass filtered recorded wind direction data, in this study, we develop a regression-based wind
direction forecast. The wind direction forecast uses two previous time windows to identify if the wind direction is stationary
or varying in time. If the wind direction is stationary, the standard filtered wind direction is used. If the wind direction is
identified to be varying in time, a linear regression is used to forecast the future wind direction. The proposed framework can
be used with arbitrary wind direction timeseries estimation methods. Future work should consider nonlinear regression or more
complex timeseries machine learning methodologies, such as recurrent neural networks.

Closed-loop wake steering control is compared to baseline yaw aligned control and open-loop lookup table control for
yaw set-point update periods of $T = 15$ and 30 minutes. Wake steering has a larger increase in energy production for stable
ABL conditions than for convective. Open-loop lookup table control and closed-loop wake steering control with deterministic
set-point optimization have reduced energy production in convective conditions compared to baseline yaw aligned control.
Closed-loop wake steering with set-point optimization under uncertainty increases energy in convective conditions, compared
to baseline control. The highest overall energy production is achieved with closed-loop wake steering with yaw misalignment
set-point optimization under wind direction and model parameter uncertainty for $T = 15$ minutes. Reducing the yaw set-point
update period increases the energy production for all cases except for closed-loop wake steering control with deterministic set-
point optimization, where the yaw set-points are influenced by data measurement noise. The influence of the data measurement
noise is alleviated with set-point optimization under uncertainty.

The results of Part 1 and 2 of this study suggest several directions of future work. Future work should investigate the optimal
yaw set-point update period in tandem with wind condition prediction methodologies. Realistic utility-scale turbine yaw duty
penalties, based on yaw motor energy usage and increased maintenance costs, should also be considered in the set-point
optimization. Improved estimates for the wake model parameter probability distributions with physical constraints should be
considered. Additionally, future work should consider model form uncertainty and modeling error in connection with model
parameter estimation. Future studies should compare various model-based closed-loop wake steering approaches which use
steady-state and dynamic wind farm models to model-free closed-loop wake steering control.

We note that the simulations presented in this study are an idealization of the diurnal ABL with fixed geostrophic wind
speed and direction (Beare et al., 2006; Svensson et al., 2011; Fitch et al., 2013). While observations occasionally demonstrate
approximately steady geostrophic winds over timescales up to a day (Bosveld et al., 2014), variations in the large-scale forcings
in the atmosphere influence the ABL (Muñoz-Esparza et al., 2017) and wind farm flows (Sanz Rodrigo et al., 2017a, b).
Methodologies to investigate wake steering control in more realistic ABL wind conditions through meso-microscale coupling
should be considered in future work. Finally, future work should consider wake steering in complex terrain.

## Appendix A: Statistical wind direction forecast algorithm

The regression-based statistical wind direction forecast discussed in §2.1 is described in this section. A schematic of the
algorithm is shown in Figure A1. The algorithm is presented in Algorithm 1. The inputs are the measurement time series $t$, the





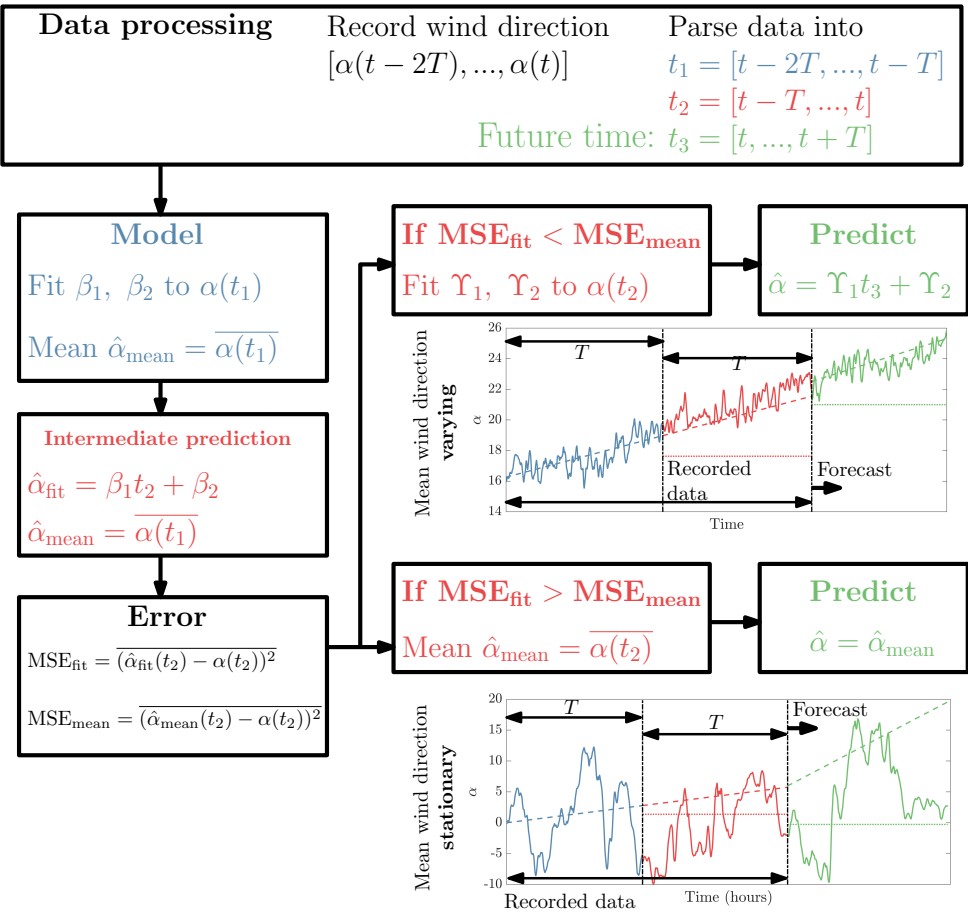

**Figure A1.** Wind direction estimation algorithm DirectionEstimation. Dashed lines are wind direction predictions using regression and dotted lines are predictions using the mean values.

measured wind direction time series $\alpha$, the current time $t_s$, the yaw set-point update period $T$, and the minimum coefficient of determination value $R_{\min}$. The algorithm Regression() is provided time and wind direction vectors and uses linear regression to estimate the wind direction over the next time period of length $T$ ($\alpha_F$). Averaging is denoted by $\langle \cdot \rangle$.

**Appendix B: Diurnal cycle validation**

The diurnal cycle implementation in *PadéOps*[1] (Ghate, 2018) is validated in this section. The diurnal cycle LES case of Kumar et al. (2006) is used as a reference. The boundary conditions constructed in Kumar et al. (2006) correspond to the HATS field campaign (Horst et al., 2004). The full details of the diurnal cycle initialization are provided in Kumar et al. (2006). The free atmosphere is in geostrophic balance. Only the vertical component of Earth's rotation is included (traditional approximation is 500 enforced (Howland et al., 2020b)). The prescribed geostrophic wind speed and surface heat flux are shown in Figure B1.



---

**Algorithm 1** Regression-based wind direction forecast with uncertainty

---

DirectionEstimation($\boldsymbol{t}, \boldsymbol{\alpha}, t_s, T, R_{\min}$):

$\boldsymbol{\alpha}_{F1},\ R_1^2 \leftarrow \text{Regression}(\boldsymbol{t}(t_s - 2T : t_s - T)),\ \boldsymbol{\alpha}(t_s - 2T : t_s - T),\ T)$

$\boldsymbol{\alpha}_{F2},\ R_2^2 \leftarrow \text{Regression}(\boldsymbol{t}(t_s - T : t_s),\ \boldsymbol{\alpha}(t_s - T : t_s),\ T)$

$\varepsilon_f \leftarrow \left\langle (\boldsymbol{\alpha}(t_s - T : t_s) - \boldsymbol{\alpha}_{F1}(t_s - T : t_s))^2 \right\rangle$

$\varepsilon_m \leftarrow \left\langle (\boldsymbol{\alpha}(t_s - T : t_s) - \langle \boldsymbol{\alpha}(t_s - 2T : t_s - T) \rangle)^2 \right\rangle$

**if** $\varepsilon_f < \varepsilon_m$ and $R_1^2 \geq R_{\min}$ and $R_2^2 \geq R_{\min}$ **then**

    $\alpha_{\text{STD}} = \text{STD}(\boldsymbol{\alpha}(t_s - T : t_s) - \boldsymbol{\alpha}_{F1}(t_s - T : t_s))$

    $\hat{\alpha} = \boldsymbol{\alpha}_{F2}(t_s + T/2)$

**else**

    $\alpha_{\text{STD}} = \text{STD}(\boldsymbol{\alpha}(t_s - T : t_s))$

    $\hat{\alpha} = \langle \boldsymbol{\alpha}(t_s - T : t_s) \rangle$

**end if**

**return** $\hat{\alpha}, \alpha_{\text{STD}}$

---

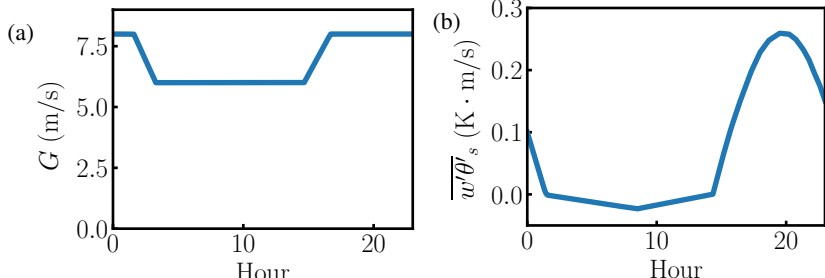

**Figure B1.** Diurnal cycle validation case (Kumar et al., 2006). (a) Diurnal cycle geostrophic wind speed. (b) Diurnal cycle surface heat flux $\overline{w'\theta'}_s$.

The validation focuses on two integrated quantities in the ABL. The friction velocity is shown in Figure B2(a). There is sufficient agreement between the present LES and the reference case. The boundary layer height $z_i$, normalized by the Obukhov length, is shown in Figure B2(b). There is qualitative agreement between the cases with some quantitative discrepancy in the stability transition regions of the profile. The quantitative discrepancies in the normalized boundary layer height are primarily the result of the differing numerics and subgrid scale models used in the two simulations. Primary discrepancies arise in the stable ABL since the Ozmidov scale is of the same order as the grid spacing (Sullivan et al., 2016). The present LES uses a $6^{th}$ order compact finite difference scheme (Lele, 1992) in the vertical direction, whereas Kumar et al. (2006) implemented a $2^{nd}$ order finite difference scheme. Overall, the results suggest that the diurnal cycle boundary condition implementation is sufficient for the simulation of a representative diurnal cycle of the ABL.





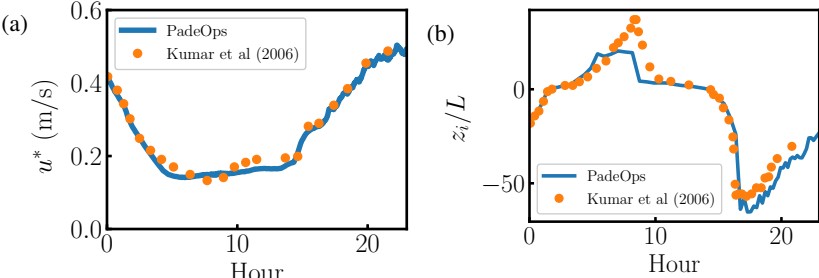

**Figure B2.** Diurnal cycle validation case (Kumar et al., 2006). (a) Diurnal cycle friction velocity $u^*$. (b) Diurnal cycle boundary layer height normalized by the Obukhov length $z_i/L$. Details for boundary layer height estimation provided in Kumar et al. (2006).

## Appendix C:  Note on wake steering LES initialization


Since turbulent flows exhibit a chaotic dependence on initial conditions (e.g. Pope, 2001), the initial conditions for all LES cases presented in this study are executed from the same initial conditions. Further, differences in parallel processor topology can result in round-off errors which will exponentially grow to $\mathcal{O}(1)$ differences in the instantaneous flow fields. In this section, we highlight the differences that arise in the comparison of separate wind farm control LES cases due to the chaotic nature

of turbulence. Two simulations of open-loop lookup table control are implemented in the diurnal cycle simulations described in §3. The lookup table methodology is described in Appendix D. The simulations are started from identical initial conditions but with different parallel processor topology, which will result in an initial round-off error difference $(10^{-8})$ between the cases. The reference turbine wind directions are shown in Figure C1(a) and the applied yaw misalignments are shown in Figure C1(b). While the differences between the cases appear minor visually, they differ in their energy ratio results. The energy

gains for the two cases with respect to baseline yaw aligned control are $3.43\%$ and $3.17\%$ for cases 1 and 2, respectively. The primary differences arise in convective ABL conditions, where the energy gains are $-0.74\%$ and $-1.69\%$ for cases 1 and 2, respectively. Conversely, the differences in stable conditions are minor, with gains of $4.70\%$ and $4.72\%$ for cases 1 and 2, respectively. Overall, the results suggest that the initialization and parallel processor topology round-off must be identical to machine precision to ensure accurate comparisons between LES control cases. Primary differences arise in ABL conditions

with high turbulence.

## Appendix D:  Lookup table synthesis

The yaw misalignment lookup table synthesis is described in this section. The wake model presented in Part 1 (Howland et al., 2020c) is used for yaw set-point optimization for the eight wind turbines of interest (Figure 2) for the wind directions encountered in the LES case, between $-10° < \alpha < 30°$ (Figure 3(a)). The wake spreading rate is prescribed using the empirical

fit of Niayifar and Porté-Agel (2016), $k^* = 0.3837 \cdot \text{TI} + 0.003678$, where TI is the streamwise turbulence intensity. The proportionality constant of the presumed Gaussian wake is set to $\sigma_0 = 0.25$ (Shapiro et al., 2019; Howland et al., 2020c). The





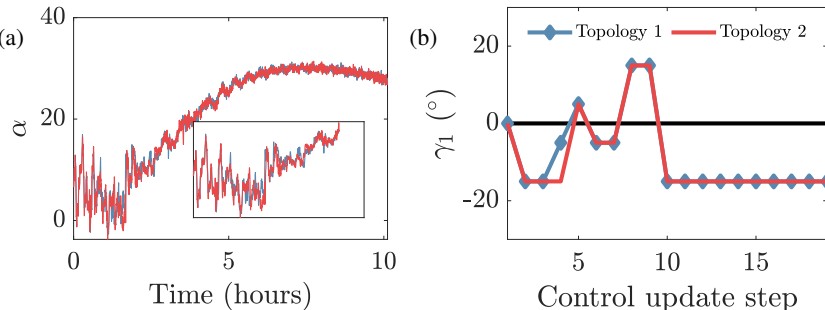

**Figure C1.** (a) Reference turbine wind direction for open-loop wake steering control cases for two differing parallel processor topologies. The simulations are executed from identical initial conditions. (b) The yaw misalignment set-points implemented in the two open-loop lookup table control cases.

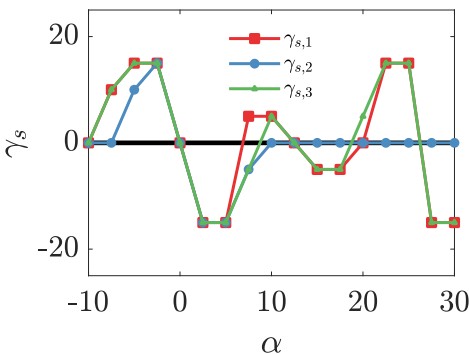

**Figure D1.** Yaw misalignment set-point lookup table for open-loop control for turbines 1, 2, and 3 as a function of the incident wind direction.

parameter $k^*$ in the empirical fit of Niayifar and Porté-Agel (2016) is not identical to the parameter $k_w$ used in the lifting line wake model (Shapiro et al., 2018). An empirical calibration for $k_w$ is not available in the literature. Instead, the wake spreading rate $k_w$ is found by equating the Gaussian wake model form used by Niayifar and Porté-Agel (2016) with the form used in this 535 study. The resulting empirical $k_w$ is

$$k_w = \frac{k^* x + 0.2\sqrt{\frac{1+\sqrt{1-C_T}}{2\sqrt{1-C_T}}} - 1}{\sigma_0 \log(1 + \exp(2(x-1)))}, \tag{D1}$$

where $x$ is the streamwise distance between the turbines normalized by the rotor diameter, $C_T$ is the coefficient of thrust, and $k^*$ is defined in the relationship above.

The yaw set-point lookup table is constructed with a wind direction discretization of $\Delta \alpha = 2.5°$. The turbulence intensity is 540 extracted from the baseline yaw aligned control simulation as a function of time. The mean turbulence intensity in each wind direction bin (see Figure 3(c)) is used to estimate the wake spreading rate $k^*$, which is then used to compute $k_w$ in Eq. D1. The yaw set-points are then optimized in each wind direction bin for the prescribed $k_w$ and $\sigma_0$. The resulting yaw set-points for turbines 1, 2, and 3 are shown in Figure D1. The other yaw misalignments are not shown for brevity, but are provided in the



dataset accompanying this study. The yaw misalignments are applied to the wind farm by selecting the closest wind direction

bin to the moving average filtered wind direction estimate (see Figure 12).

*Author contributions.* M.F.H. and J.O.D. conceived the work. A.S.G. and M.F.H. developed the LES code. M.F.H. conducted analysis. M.F.H. wrote the manuscript. All authors contributed to edits.

*Competing interests.* The authors declare no conflicts of interest.

*Acknowledgements.* A.S.G. was funded by Tomkat Center for Sustainable Energy at Stanford University. S.K.L. acknowledges partial sup-
port from NSF-CBET-1803378. M.F.H. acknowledges partial support from Siemens Gamesa Renewable Energy. All simulations were per-
formed on Stampede2 supercomputer under the XSEDE project ATM170028.

*Code and data availability.* All data and computer code used in this paper is open source. The data and code are available at: https://doi.org/
10.5281/zenodo.5160943 (Howland, 2021a).



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
