# Peer review of "Optimal closed-loop wake steering, Part 2: Diurnal cycle atmospheric boundary layer conditions"

_Wind Energy Science, 2021_

## Referee Comment (RC2)

**Review of *Optimal closed-loop wake steering, Part 2: Diurnal cycle atmospheric boundary layer conditions* by Michael F. Howland et al.**

Reviewer: M. Paul van der Laan, DTU Wind Energy

The authors investigate the benefits of different wake steering control methods to enhanced the power production in a wind farm subjected to a transient atmospheric boundary layer using large-eddy simulations. The article is well written and I only have a small number of comments. Note that I am not a control expert and I have chosen to focus on the overall methods. I have listed main and minor comments below, which should be addressed in order to accept the article as a publication for Wind Energy Science.

**Main comments**

1. Line 50: *Stable ABL low-level jets are generated, in part, by inertial oscillations induced by Coriolis forces (Van de Wiel et al., 2010).* The mean effect of the low-level jet can be explained in a more simple way see a recent work of the reviewer van der Laan et al. (2021), where it is shown that the Coriolis-induced wind veer causes the jet. Once the wind veer is removed, using the veer-less ABL model, then the jet is neither present. This is the case for analytic solutions, RANS models and LES. (van der Laan et al. (2021) does not include LES results but a colleague has tested the veerless model in an unpublished work using the GABLS test case from Svensson et al. (2011) and neither got a jet.)

2. Figure 2: What is the spacing between the wind turbines? You should mention this important parameter in the text or you could plot the layout normalized by the rotor diameter and use grid line in the plot. It looks like the chosen spacing between the wind turbines in $y$ is relatively small (4D?), which enhances the benefits of wake steering. I think you should note that you are investigating a relative small spacing and that your energy gains due to wake steering are expected to be less for large wind turbine spacing. Why didn't you use a more realistic wind farm layout representing a modern wind farm? In other words, how would your main conclusions on yaw control methods change for different wind farm layouts?

3. Wind direction and stability: Do I understand correctly that you are both investigating the effect of stability and wind direction at the same time and that stable and unstable conditions reflect South-Western and Western wind directions, respectively? If yes, then it is not fair to compare stable with unstable directly (as you do in the paper in lines 340-342, Table 1 and elsewhere) because they represent different wind direction flow cases. If you want to compare stable with unstable, you should have the same wind direction or you could perform multiple wind directions for both stable and unstable such that your model results represent the same wind rose. If the latter is not desired by authors then you should at least rename and clarify the cases (to for example something like stable-SW and unstable-W or stable-diagonal and unstable-row). If the wind direction is the main parameter of interest for the yaw control optimization studies, then you could have used a quasi-steady stable and unstable ABL using a geostrophic wind direction that varies over time (in order to get the same wind direction cases).

4. Section 4.1 and Figure 6: Do I understand correctly that you derive a power-yaw relationship for the two leading wind turbines using the LES results where all wind turbines are active? This can lead to different results compared a wind turbine in isolation, which is normally used to estimate a power-yaw relationship. I think you should mention this in the article.

5. Appendix C: Note on wake steering LES initialization: You mention that you cannot get the same result by running the same LES on a different number of CPUs. First if all, I appreciate the fact that you also report the model challenges. I agree that the chaotic nature of the real atmospheric makes it impossible to measure the same ABL twice. However, for idealized CFD simulations of the ABL, one should be able to get the same result regardless of the number of CPUs, as long as the number of CPUs does not affect the number of cells in the numerical grid, the use of random number generators is avoided (or used with a fixed seed) and the communication between CPUs is handled in a consistent manner. The LES model of the in-house CFD (finite volume) code of DTU Wind Energy (EllipSys3D) does currently not have the parallelization issue. However, we have noticed that round off issues due to inconsistent communication between CPU's previously led to a different times at which the turbulence started to form (which has been fixed). The authors are welcome to contact the reviewer (after the review process) for further discussion.

**Minor comments**

1. Section 3: You could adde that you use a barotropic atmosphere since you use a constant geostrophic wind speed.

2. Figure 5, caption: Hub height velocity should be hub height wind speed (or stream-wise velocity?).

**References**

Svensson, G., Holtslag, A. A. M., Kumar, V., Mauritsen, T., Steeneveld, G. J., Angevine, W. M.and Bazile, E., Beljaars, A., de Bruijn, E. I. F., Cheng, A., Conangla, L., Cuxart, J., Ek, M., Falk, M. J., Freedman, F., Kitagawa, H., Larson, V. E., A., L., Mailhot, J., Masson, V., Park, S., Pleim, J., Söderberg, S., Weng, W., and Zampieri, M.: Evaluation of the Diurnal Cycle in the Atmospheric Boundary Layer Over Land as Represented by a Variety of Single-Column Models: The Second GABLS Experiment, Boundary-Layer Meteorology, 140, 177–206, https://doi.org/10.1007/s10546-011-9611-7, 2011.

van der Laan, M. P., Kelly, M., and Baungaard, M.: A pressure-driven atmospheric boundary layer model satisfying Rossby- and Reynolds number similarity, Wind Energy Science Discussions, 2021, https://doi.org/10.5194/wes-2020-130, 2021.

---

## Author Comment (AC1)

**Response to Referee #1**

We thank the referee for their review and their thoughtful comments. Point-to-point responses can be found below, and the relevant changes will be made to the manuscript during the revised manuscript submission stage.

**Main comments:**

**Comment #1**
I sincerely thank you for your invaluable contributions to the research field. Your manuscript contains notable and novel contributions that further the state of the art in wind farm control. I find the scientific relevance and level of detail both excellent. I very much enjoyed the thorough literature review in the introduction, and how this article combines both a high-quality large-eddy simulation study with an advanced wind farm control solution. Important findings regarding wind direction forecasting, real-time model parameter estimation and yaw optimization is likely to shape the next step in wake steering after open-loop control. Generally, the paper is dense in information but does not go too far in that matter. I only have a handful of minor remarks.

**Response**
Thank you for your thoughtful feedback. We appreciate your insights and provoking questions which will help guide the present and future work.

**Point comments:**

**Comment #1**
Can you say more about the inherent assumptions in Equation (1)? Are you assuming wakes to propagate instantly? Is there anything you can do to include time dependency in this optimization?

**Response**
In our approach, we use a steady-state wake model to optimize the yaw set-points. This inherently assumes that the model-optimal yaw set-points resulting from maximizing power in the wake model are appropriate yaw set-points to use in the wind farm LES. Given the steady-state wake model, the primary assumption in Eq. (1) is that the flow is statistically steady-state from $t_0$ to $t_0 + T$, where $t_0$ is the current time and $T$ is the yaw update period. This inherently neglects the time delay associated with the yaw maneuver (i.e. from the current yaw state to the yaw state $\gamma_s$ resulting from Eq. (1) [see ref. 1]). It also neglects the energetic cost associated with the yaw motor actuation. We assume that wind direction variations over $t_0$ to $t_0 + T$ can be accounted for with $f(\alpha)$. We have added more discussion of the inherent assumptions in Eq. (1) to the manuscript.

**Comment #2**
Equation (2), near line 150, you state that the wind direction is assumed to be uniformly distributed. In other work, the wind direction is often assumed to have a Gaussian distribution (Rott et al., Simley et al.) or a Laplace distribution (Quick et al.). Could you explain your choice?

**Response**
Thank you for noting this. Yes, previous authors, Rott et al., Simley et al., and Quick et al used Gaussian or Laplacian distributions to describe variations of the wind direction about a statistically stationary

mean. We used uniform distributions for $f(\alpha)$ primarily because the wind direction is statistically non-stationary in our simulations (see e.g. Figure 3 in the paper). For a wind direction which is evolving in time (e.g. over the window $t_0$ to $t_0 + T$), assuming a normal distribution about a stationary mean state may underpredict the probability of deviations from the mean state. The best choice of $f(\alpha)$ will also inherently depend on T in transient flow. For smaller values of T, a normal distribution is more appropriate, as shown by previous authors (e.g. 5 minute window in Rott et al). Using a uniform distribution is likely not the best choice. We recommend future work to investigate the appropriate representation of $f(\alpha)$ in evolving ABL conditions with mesoscale and diurnal cycle driven non-stationarity. We have added more discussion to the manuscript.

**Comment #3**

Line 177: "The standard approach … direction filter used." You mention a "low-pass moving averaged filtered wind direction". A moving average filter also falls within the class of lowpass filters, as far as I understand it. The way you specify it; do you mean that you used two filters, one to lowpass filter the signal and one to additionally calculate a moving average from those filtered values? Similarly, you cite Simley et al. (2020) to use a "first order filter", but this is also a lowpass filter. You state that the results do not really change much based on which filter is used, and I assume you are already aware of the things I stated here, but this paragraph was not completely clear for me.

**Response**

Thank you for this note, we apologize that the paragraph was confusing to read and we have modified the language to improve clarity. Yes, a moving average is a type of low-pass filter, as is the first order filter in Simley et al. (2020). We do not include an additional low pass filter beyond the moving average.

**Comment #4**

Line 185, top of page 7: I really appreciate the simplicity of assuming a linear trend for the wind direction, if a certain threshold is met. Is there a theoretical motion that would support the decision to model this as a linear process?

**Response**

To the best of the authors knowledge, there is not a justification to model the wind direction trend as a linear process other than the empirical results of the present diurnal cycle simulation. We recommend further investigation of short-term wind direction forecasts in future work. Such methods could include either physics- (based on the momentum equations) or data-driven approaches. We have added discussion of this topic to the paper.

**Comment #5**

Line 195, Section 3: To de-condense the text somewhat, please consider putting important details of the LES simulation in a table.

**Response**

We have put the simulation details in a table.

**Comment #6**

Line 214: Perhaps remove footnote 1 and instead add an entry to the reference list.

**Response**

We have added the code *PadeOps* as a reference item.

**Comment #7**

Page 8, Figure 1, and lines 219 until 227: I was wondering if this information is essential in the main text. I do see the value of explaining why and how the wind direction changes, but perhaps it is not essential to the story you are trying to tell in this article. Showing figure 3 should provide the reader with sufficient information in how the ambient conditions will change and under what conditions the turbines and the wind farm controllers are subjected. Perhaps some of this information can be moved to an appendix.

**Response**

Thank you for this comment. We have moved Figure 1 to the appendix as the referee has suggested. I appreciate the need to write concisely to not complicate the primary narrative of the paper. However, I believe this short discussion regarding the wind condition variations is useful information to understand the character of the atmospheric boundary layer conditions that the wind farm will experience and to provide physical intuition for the time-varying nature of the flow. I hope that this discussion will help to elucidate the physical mechanisms which cause ABL variations to which the controller reacts in the results section.

**Comment #8**

Figure 4: I think this figure is really interesting. You may consider it moving to an appendix, as stated in the previous remark. Also, could you please add a legend defining the time window for each vertical line.

**Response**

We have added a legend specifying the time window for each vertical line. As with **Comment #7**, we believe the nature of the wind conditions and their variation in time are of importance to the wind farm control strategy and also the interpretation of the results.

**Comment #9**

Figure 5: this is a very informative figure. It is a little difficult to see at the current resolution. Could you perhaps update the xlims/ylims, zooming in to the region of interest? Also, if can consider removing the yticks and ylabels from subplots (b) and (d), and similarly for the x-axis for plots (a) and (b).

**Response**

We have added a zoomed figure in the Appendix. I prefer not to put only the zoomed figure in the main text as this may confuse readers into thinking that is our full computational domain, but it is important readers are able to see the details in the zoomed figure, so we have added it to the Appendix.

**Comment #10**

Line 225, Section 4: After reading "Case A" (and "Case D" later on), I was expecting to also see a "Case B" and "Case C". I later realized what they were actually supposed to mean. To clarify this and also to de-condense the text somewhat, please add a table defining the various cases. I think with that table,

you can keep the case naming convention you have now. A table would really make it easier for the reader to see what cases were tested and what combinations of controllers, prediction vs. past-time window- averaged wind direction estimates, with and without uncertainty.

**Response**

We have added a table describing each case.

**Comment #11**

Line 307: The default cases are presented with a control update period of T=30 minutes. This seems very high to me, especially if you are anticipating a change in the mean wind direction. It there a reason you picked such a high value to start with?

**Response**

Thanks for this question. The timescale with which the yaw misalignment should be updated inherently depends on the incident wind conditions and the timescales of their variability. We have submitted a separate paper to *ACC* to investigate the influence of the stability in the ABL on the best yaw update period T, which was not the primary focus of the present study. In the present study, we investigated T=15 min and 30 min. In the present idealized diurnal cycle simulations, the variation of the mean wind conditions are relatively slow (diurnal timescales) and a longer update period is more justified. Future work should investigate the influence of mesoscale structures in the atmosphere (not present in microscale LES) on the optimal update period T.

**Comment #12**

Page 14: explanations are very clear, really excellent.

**Response**

We thank the referee for this comment.

**Comment #13**

Figure 9: plots are somewhat small, while the xlabel and ylabel are large. Could you enlarge the actual plots? Also, could you export these plots in vectorized format (.pdf, .eps) so that I can zoom in at a high resolution?

**Response**

We have enlarged the plot in the revised manuscript. The plots are in vectorized format (.eps) which I believe should be available when the paper is posted online? If not, please reach out to me at mhowland@mit.edu for the plots. Alternatively, the dataset is already posted online here: https://doi.org/10.5281/zenodo.5160943

**Comment #14**

Figure 10: Can the EnKF estimate the model parameters if there is no wake interaction? Will your estimates drift off if you are without wake interaction for an extended period of time?

**Response**

We have a simple geometry-based wake detection algorithm (based on the wake diameter and wake expansion as a function of downwind distance) that avoids calculating wake contributions when there are no interactions for reduced computational effort. So our EnKF does not estimate parameters if there

is no wake interaction. Regarding the details of the EnKF if we did not have this flag, the wake model parameters would not be statistically altered if there were no wake interactions because the power prediction deviation matrix $\widehat{\Pi}' = 0$ (see Ref [2]).

**Comment #15**
Line 356: "The wake model… exhibit low error." If you use power production measurements in your EnKF, is the model-predicted power production a fair value to use in validation? Had you calibrated your EnKF to have a much lower measurement noise covariance matrix than the process noise cov. matrix, then would you not only further improve this quantity?
**Response**
The referee is exactly correct that this is simply a statement that the calibration (training) error is low and that it could be even further reduced. We did not intend this statement to reflect a validation of the EnKF approach. In fact, this is the focus of Figure 12 where we compared wake model predictions (out-of-sample) to the LES output. The EnKF reduces the error of the predictions (out-of-sample) in addition to the reduction of the calibration error. We have further clarified this line in the manuscript.

**Comment #16**
How large is the sample pool in the EnKF, and what did you base this on?
**Response**
I assume the referee is referring to the number of ensemble members, for which we use 100, a commonly used number [3] which we also used in Part 1 [2]. We selected 100 based on offline hyperparameter tuning experiments (see Ref. [2]).

**Comment #17**
Figure 12 says "Case SF", but no such case was introduced in the text. Did you mean OOU-F?
**Response**
Thank you for catching this typographical error. We have fixed it.

**Comment #18**
I would very much like to see you separate the update/averaging sampling time for the model estimation part from the control setpoint update rate. Naturally, I can see that you may not need to update the model parameters very frequently, notably since perhaps the wake expansion parameter need not change very fast. However, the optimal yaw setpoint may need to change much at a much higher frequency. I think there still may be a lot to gain here. You rightfully mention it in your text, but I wanted to emphasize that I am very curious to see how your results would change. Again, this is not something I expect you to address in this manuscript.
**Response**
Thank you for this comment. I agree entirely that it is likely the parameters could be updated with a slower frequency than the yaw, and that we can/should decouple these two periods. We have begun an investigation in this direction, and we anticipate investigating it more thoroughly in future work. There are several technical and logistical challenges which need to be overcome to enable this decoupling.

Future work should primarily consider what are the atmospheric determinants which affect the wake spreading rate to identify with what frequency they should be updated.

**Comment #19**

Line 448-465: I wonder if these two paragraphs could be omitted. They seem to be a general summary of the methodology and background information. I think you can assume that the reader has read at the very least the introduction of your manuscript. This would make the manuscript's results and conclusions stand out more.

**Response**

Thanks for this comment. We have removed the first paragraph from the conclusions to, as the referee has suggested, allow the conclusions and discussion stand out.

References

[1] Macrí, Stefano, Sandrine Aubrun, Annie Leroy, and Nicolas Girard. "Experimental investigation of wind turbine wake and load dynamics during yaw maneuvers." *Wind Energy Science* 6, no. 2 (2021): 585-599.

[2] Howland, Michael F., Aditya S. Ghate, Sanjiva K. Lele, and John O. Dabiri. "Optimal closed-loop wake steering–Part 1: Conventionally neutral atmospheric boundary layer conditions." *Wind Energy Science* 5, no. 4 (2020): 1315-1338.

[3] Howland, Michael F., Oliver RA Dunbar, and Tapio Schneider. "Parameter uncertainty quantification in an idealized GCM with a seasonal cycle." *arXiv preprint arXiv:2108.00827* (2021).

---

## Author Comment (AC2)

**Response to Referee #2**

We thank the referee for their review and their thoughtful comments. Point-to-point responses can be found below, and the relevant changes will be made to the manuscript during the revised manuscript submission stage.

**Main comments:**

**Comment #1**
Line 50: *Stable ABL low-level jets are generated, in part, by inertial oscillations induced by Coriolis forces (Van de Wiel et al., 2010)*. The mean effect of the low-level jet can be explained in a more simple way see a recent work of the reviewer van der Laan et al. (2021), where it is shown that the Coriolis-induced wind veer causes the jet. Once the wind veer is removed, using the veer-less ABL model, then the jet is neither present. This is the case for analytic solutions, RANS models and LES. (van der Laan et al. (2021) does not include LES results but a colleague has tested the veerless model in an unpublished work using the GABLS test case from Svensson et al. (2011) and neither got a jet.)
**Response**
Thank you for the comment and for directing us to van der Laan et al. (2021). This is a great paper, and provides useful insight using idealized models.
*Modification:*
Stable ABL low-level jets are generated, in part, by Coriolis-induced wind veer (van der Laan et al. (2021)) and by inertial oscillations induced by Coriolis forces (Van de Wiel et al., 2010).

**Comment #2**
Figure 2: What is the spacing between the wind turbines? You should mention this important parameter in the text or you could plot the layout normalized by the rotor diameter and use grid line in the plot. It looks like the chosen spacing between the wind turbines in y is relatively small (4D?), which enhances the benefits of wake steering. I think you should note that you are investigating a relative small spacing and that your energy gains due to wake steering are expected to be less for large wind turbine spacing. Why didn't you use a more realistic wind farm layout representing a modern wind farm? In other words, how would your main conclusions on yaw control methods change for different wind farm layouts?
**Response**
Thanks for highlighting this important point. We have added the spacing to Figure 1. The referee is correct that the gains from wake steering are specific to the wind farm geometric setup among other details of the simulation, including but not limited to the turbulence in the ABL, the surface heat flux boundary condition, the geostrophic wind speed and direction, and the shear/veer. We selected the layout of the farm to be a representative, idealized test case for wind farm control similar to previous wind farm control studies [1] and with spacing comparable to wind field sites of interest from our research [2]. We have added further discussion to highlight that the gains from wind farm control will be case specific.

**Comment #3**
Wind direction and stability: Do I understand correctly that you are both investigating the effect of stability and wind direction at the same time and that stable and unstable conditions reflect South-Western and Western wind directions, respectively? If yes, then it is not fair to compare stable with unstable directly (as you do in the paper in lines 340-342, Table 1 and elsewhere) because they represent different wind direction flow cases. If you want to compare stable with unstable, you should have the same wind direction or you could perform multiple wind directions for both stable and unstable

such that your model results represent the same wind rose. If the latter is not desired by authors then you should at least rename and clarify the cases (to for example something like stable-SW and unstable-W or stable-diagonal and unstable-row). If the wind direction is the main parameter of interest for the yaw control optimization studies, then you could have used a quasi-steady stable and unstable ABL using a geostrophic wind direction that varies over time (in order to get the same wind direction cases).

**Response**

The primary purpose of the paper is to test the closed-loop wake steering methodology proposed in Part 1 (Howland *et al., Wind Energy Science*, 2020, 5, 1315-1338) in an idealized ABL case with inflow wind condition variations as a function of time. In Part 1, we develop a closed-loop wake steering methodology and test its performance in the statistically quasi-stationary CNBL. Here, we are interested in comparing the performance of the closed-loop control to existing open-loop control methodologies in an example transient ABL case. We select the diurnal cycle case to exhibit both wind direction and stability variations, which both have substantial impact on wake steering control.

The referee is correct that the present simulation does not contain controlled experiments between stable and unstable cases since the wind direction has changed. We have clarified this further in the manuscript. It is worth noting that the wind turbine spacing is larger for the stable-SW condition than for the unstable-W condition. Given the turbine spacing and also the multiple wake interactions superposed, for fixed stability, the flow from the west would likely result in larger wake losses and likely more potential for wake steering. However, the effect of stability outweighs the farm geometry in this particular simulation.

We have added more discussion of this point to the paper and modified Table 1 to note the wind direction.

**Comment #4**

Section 4.1 and Figure 6: Do I understand correctly that you derive a power-yaw relationship for the two leading wind turbines using the LES results where all wind turbines are active? This can lead to different results compared a wind turbine in isolation, which is normally used to estimate a power-yaw relationship. I think you should mention this in the article.

**Response**

We agree that this could lead to different results compared to a wind turbine in isolation. We have added further discussion to the manuscript on this point.

**Comment #5**

Appendix C: Note on wake steering LES initialization: You mention that you cannot get the same result by running the same LES on a different number of CPUs. First of all, I appreciate the fact that you also report the model challenges. I agree that the chaotic nature of the real atmospheric makes it impossible to measure the same ABL twice. However, for idealized CFD simulations of the ABL, one should be able to get the same result regardless of the number of CPUs, as long as the number of CPUs does not affect the number of cells in the numerical grid, the use of random number generators is avoided (or used with a fixed seed) and the communication between CPUs is handled in a consistent manner. The LES model of the in-house CFD (finite volume) code of DTU Wind Energy (EllipSys3D) does currently not have the parallelization issue. However, we have noticed that round off issues due to inconsistent communication between CPU's previously led to a different times at which the turbulence started to form (which has been fixed). The authors are welcome to contact the reviewer (after the review process) for further discussion.

**Response**

Our in-house LES code *PadeOps* (https://github.com/FPAL-Stanford-University/PadeOps) is pseudo-spectral (spectral methods in horizontal, 6th order finite difference in vertical direction). As the referee has noted, small differences in two solutions at round-off magnitude grow exponentially in time due to the instability of the chaotic Navier-Stokes system (Lyapunov stability). In our experience with wall-bounded turbulence, round off errors of the order 1E-13 generate order 1 deviations in local skin frictions in about 250-500 RK4 time steps.

There are several aspects of the discretization algorithm that introduce these round-off level perturbations. The spectral discretization requires Fourier transforms that are very aggressively optimized for the specific decompositions (for example the reviewer is referred to the "exhaustive" plans available as part of the FFTW library for complex Fourrier transforms to identify efficient algorithms in routines such as "fftw_plan_many_dft"). This aggressive optimization ensures that the most optimal FFT algorithm is identified and used for each specific individual simulation. This is the primary source of round-off perturbations at double-precision level determined by the compiler type, optimization flags and distributed memory partitioning topology.

As discussed in Appendix D, round-off error is introduced by changing this parallel processor topology. While these round-off errors grow to order 1 deviations in instantaneous snapshots, it is not expected that they would alter mean flow statistics (averaged over long periods). However, in the control simulations presented, which use Fourier collocation, the wake steering controller interacts in a nonlinear fashion with finite time-averaged statistics, and differences in the finite time-averaged power production statistics appear (shown in Appendix D). When we fix the parallel processor topology, round-off error is eliminated in *PadeOps* (Appendix D). We would also be happy to discuss the DTU code's methodology in the future to hear about the solution methods which have been implemented.

**Point comments:**

**Comment #1**

Section 3: You could adde that you use a barotropic atmosphere since you use a constant geostrophic wind speed.

**Response**

We have added that we consider a barotropic atmosphere with no geostrophic wind shear from baroclinicity.

**Comment #2**

Figure 5, caption: Hub height velocity should be hub height wind speed (or stream-wise velocity?).

**Response**

We have clarified this in the manuscript.

References

[1] Doekemeijer, Bart M., Daan van der Hoek, and Jan-Willem van Wingerden. "Closed-loop model-based wind farm control using FLORIS under time-varying inflow conditions." *Renewable Energy* 156 (2020): 719-730.

[2] Michael F. Howland, "Wind farm yaw control set-point optimization under model parameter uncertainty", Journal of Renewable and Sustainable Energy 13, 043303 (2021) https://doi.org/10.1063/5.0051071

---

## Author Response (AR2)

**Response to Associate Editor**

We thank the Associate Editor for their review and their thoughtful comments. Point-to-point responses can be found below, and the relevant changes have been made to the manuscript.

**Comments:**

**Comment #1**

The gains (as with most of these studies) are quite sensitive to the particular set-up - number of turbines, spacing, flow cases, etc - and this study only considered one such case. This is fine because the goal is to demonstrate the method and its performance under different stability conditions (and this is addressed), but it would be good to expound a bit on the limitations from this (i.e. in the abstract, intro and conclusions)

**Response**

Thank you for this comment. We agree that the gains from wake steering will be case specific, and depend on the wind farm design, in addition to the atmospheric flow conditions. We have added more discussion of this to the manuscript to highlight this point.

**Comment #2**

It is good that there was a brief mention of yaw duty cycle in the conclusions, but there could be more discussion around loading. One thing that is immediately apparent in figure 11 is the increased cycling of yaw set point which will obviously affect loads not just of the yaw bearing and wear on the actuators but other components as well. It is interesting to note that the open loop approach does quite well actually and with less frequent updates of the yaw set points. It would be good to call that out as a potential limitation of the method. In recent work, in fact, Fleming and others have explored even more basic strategies (fixed yaw offsets) than even open loop. I would like to see a bit more discussion on this in the results and conclusions section.

**Response**

This is a very good point and relates to areas of future work for us. The results in this paper indicate that the yaw duty may increase when using the closed-loop control method, compared to open-loop, when the closed-loop control uses power production maximization as the objective function. Our closed-loop framework can also be extended to explicitly incorporate yaw duty cost in the objective function. For open-loop wake steering, the yaw duty is a function of the objective function in the lookup table bin, but also depends on how the lookup table is applied. We have added more discussion of this to the manuscript.

[revised manuscript text omitted]